

# Mobile-assisted and gamification-based language learning: a systematic literature review

Kashif Ishaq[1], Nor Azan Mat Zin[1], Fadhilah Rosdi[1], Muhammad Jehanghir[2], Samia Ishaq[3] and Adnan Abid[4]

[1] Faculty of Information Science and Technology, Universiti Kebangsaan Malaysia, Bangi, Malaysia
[2] Institute of Education and Research, University of the Punjab, Lahore, Punjab, Pakistan
[3] School Education Department, Sheikhupura, Pakistan
[4] Department of Computer Science, University of Management & Technology, Lahore, Lahore, Pakistan

Corresponding author
Adnan Abid,
adnanabid7@gmail.com

## ABSTRACT

Learning a new language is a challenging task. In many countries, students are encouraged to learn an international language at school level. In particular, English is the most widely used international language and is being taught at the school level in many countries. The ubiquity and accessibility of smartphones combined with the recent developments in mobile application and gamification in teaching and training have paved the way for experimenting with language learning using mobile phones. This article presents a systematic literature review of the published research work in mobile-assisted language learning. To this end, more than 60 relevant primary studies which have been published in well-reputed venues have been selected for further analysis. The detailed analysis reveals that researchers developed many different simple and gamified mobile applications for learning languages based on various theories, frameworks, and advanced tools. Furthermore, the study also analyses how different applications have been evaluated and tested at different educational levels using different experimental settings while incorporating a variety of evaluation measures. Lastly, a taxonomy has been proposed for the research work in mobile-assisted language learning, which is followed by promising future research challenges in this domain.

## INTRODUCTION

Mobile technology developments are quickly expanding the field of learning in non-formal education areas by rendering universal and instance-oriented access to privileged digital resources (*Cheon et al., 2012*). Mobile learning (m-learning) technology requires mobile devices to improve learning and academic performance by having the opportunity to learn remotely at all times in compliance with students' comforts. There have been many advantages to m-learning, including cost reductions, ubiquitous communication, research assistance, and location-based services. The goal of m-learning is to put the educational sector and associations at the center of academic progress to satisfy the users' demand for

flexibility and ubiquity (*Ishaq et al., 2020d*). Mobile devices are mostly used in developed countries, particularly for language learning purposes. Specifically, Mobile-Assisted Language Learning (MALL) implies mobile phones in the learning and teaching of languages. The mobile phone allows pupils to learn quickly to develop their language comprehension skills. Besides, a significant improvement in pedagogical methods was brought about by integrating smartphone apps and games with the curriculum, thus enabling the students to learn freely in time, space and motivation on an individual basis (*Ishaq et al., 2019*).

A significant trend in mobile learning apps development involves gamification concepts that incorporates play and fun elements to inspire and attract the learner, generally referred to as serious games. A serious game's key objective is to accomplish a learning objective in a fun mode, whereby the locus of control is with the learner (*Sandberg, Maris & Geus, 2011*). Currently, schooling is not limited to a single life stage and not exclusively in traditional education institutions. Children should not only study at school but also informally outside of school. Their casual reactions outside of the classroom provide an almost as valuable learning experience as the classrooms' organized learning environment. The integration of multimedia learning content enables learners to access appropriate information within and outside the school (*Sandberg, Maris & Geus, 2011*).

The previous review studies in the MALL domain mostly concentrated on technology-based learning and handheld devices, while less on the research frameworks, content, learning, and teaching resources, as shown in Table 1. This table compares current measures based on five essential viewpoints: targeted digital repositories, teaching and learning methods, quality assessment evaluation, research framework, and learning material. Only quality articles were reviewed published in quality journals (except workshops and seminars) and performed quality assessments by discussing research frameworks, content, and teaching and learning tools.

This article was structured as follows: "Literature Review" presented a review of relevant literature. "Research Methodology" presented the methodology adopted to perform this study, along with questions and objectives, whereas "Assessment and Discussion of Research Questions" identified and summarized answers to specific study questions. "Discussion and Future Directions" presented a blend of the discussed research by defining its taxonomy, while "Conclusion" concluded this article.

## LITERATURE REVIEW

Most of the surveys and systematically reviewed on MALL do not cover publication channels (Books or Scientific Journals), quality assessment, frameworks/model used, mobile and gamified applications used for teaching and learning, and comparison of these applications. Also, the focus was more on higher education students than on primary education. A more recent systematic review on the usage of mobile devices for language learning evaluated limited studies and from restricted repositories (*Mahdi, 2018*) (*Cho et al., 2018*). The author reviewed 20 studies for mobile devices' effect on students' achievement in which student's vocabulary learning results using handheld devices were compared to those using conventional learning (*Cho et al., 2018*).

**Table 1 Related work comparison.**

| Ref. | Title survey | Survey approach | Quality assessment | Research framework | Teaching and learning tools | Content | Targeted digital repositories |
|------|-------------|-----------------|-------------------|-------------------|----------------------------|---------|------------------------------|
| (*Hwang & Tsai, 2011*) | Research trends and Prediction of technology-based learning with the latest technology (i.e., devices or concepts) | Informal | × | × | √ | × | SSCI |
| (*Burston, 2013*) | MOBILE-ASSISTED LANGUAGE LEARNING: A SELECTED ANNOTATED BIBLIOGRAPHY OF IMPLEMENTATION STUDIES 1994-2012 | Informal | × | × | √ | √ | Google Scholar |
| (*Liu et al., 2014*) | A Look at Research on Mobile Learning in K–12 Education From 2007 to the Present | Systematic Search | × | × | √ | √ | 15 Journals |
| (*Sung, Chang & Liu, 2016*) | The effects of integrating mobile devices with teaching and learning on students' learning performance: A meta-analysis and research synthesis | Systematic Search | × | × | √ | × | ERIC, EBSCOhost, PsycINFO, JSTOR, and ProQuest |
| (*Mahdi, 2018*) | Effectiveness of Mobile Devices on Vocabulary Learning: A Meta-Analysis | Systematic Search and Snowballing | × | × | √ | × | ERIC, IEEE Xplore, IGI, Proquest, Sage, ScienceDirect, and Springer Link |
| (*Cho et al., 2018*) | The Effects of Using Mobile Devices on Student Achievement in Language Learning: A Meta-Analysis | Systematic Search and Snowballing | × | √ | × | × | ERIC, SSCI |
| This Study | Mobile-Assisted and Gamification-based Language Learning: A Systematic Literature Review | Systematic Search, Snowballing, and Quality Assessment | √ | √ | √ | √ | WoS Core Collection (High quality, more than ten rep.) |

In another study, the effects of embedded portable devices in learning and teaching were examined by reviewing 110 experimental and quasi-experimental journal papers published in 1993–2013. For the usage of mobile devices in school, a moderate mean efficiency was 0.523. *Sung, Chang & Liu (2016)* analyzed the impact size of moderators and the benefits and drawbacks of mobile learning, then synthesized based on the descriptive analysis from individual experiments at various levels of moderator variables. Another SLR of researches from 2007 to the present was on mobile education in K–12 in which (*Liu et al., 2014*) reviewed a maximum of 63 articles from 15 journals, mostly exploratory and concentrated on the educational facilities associated with smartphone usage in learning. Furthermore, patterns and critical problems are also discussed for future research. *Burston (2013)* summarized 345 MALL research studies from 1994 to 2012, in a short overview of 80 words, to encourage researchers by presenting their historical background. The analysis included the home country, first or second or foreign language, the technologies employed for mobile apps, targeted research areas, type of students, demographic, study time, and the outcome summary.

Finally, (*Hwang & Tsai, 2011*) targeted influential journals from 2001 to 2010 to investigate mobile and ubiquitous learning researches related to enhanced learning technology. The publications included Educational Technology & Society, Innovations in Education and Teaching International, Journal of Computer Assisted Learning, Computers and Education, Educational Technology Research & Development, and British Education Technology Journal. The researcher presented information for many journals, a selected search sample (primary school, secondary school, tertiary education, instructors and employees), study fields (language and arts, engineering, science, math, social sciences, and more), and countries involved. However, none focused on quality assessment, framework/models, grade or education level, adopted content, approaches, statistical analyses, and comparisons between MALL and gamified apps. In our review, both areas were thoroughly discussed and differentiated from the above studies then systematically chose methods and coded them with standard naming, according to strict guidelines.

This Systematic Literature Review (SLR) discusses MALL—learners in-depth, mobile, and game-based languages, and involves the five perspectives as in Table 1. Based on the structural analysis criteria, 67 research articles have been finalized and assessed in quantitative and qualitative terms for further analysis. The SLR's significance presents the new classification criteria, MALL research trends, developed/adopted research models, learning and teaching methods, learning content, comparisons of mobile and game-based apps, research methodologies, and approaches used to evaluate the studies. This SLR may allow instructors to build a standardized MALL environment with learning and teaching apps, learning materials, frameworks, and relevant methodologies.

The study of *Sung, Chang & Liu (2016)* chose experimental and quasi-experimental journal articles from the period 1993–2013 from ERIC and SSCI repositories (only eight journals were selected) to know the use of mobile experimental studies of teaching and learning achievement of students through these devices. The study *Hwang & Tsai (2011)* published in 2011, in which journal articles were selected from SSCI repository of the 2001–2010 period to know the status of mobile and ubiquitous learning, as well as research sample group along with learning domains related to technology, were adopted in selected articles. In the study of *Liu et al. (2014)*, the selected articles were taken from 2007 to 2012 for teaching and learning of K-12 education. The study aimed to know the effectiveness and trends of mobile devices in K-12 education, and the participants were younger than 18 years.

The study of *Burston (2013)* selected the studies from the period of 1994 to 2012 in the area of MALL and annotation in which only 80 words summary was provided consisted of country, native language, mobile technology used, learning area, type of learners, and numbers, and an overview of results. The study *Mahdi (2018)* examined the effect of vocabulary learning using mobile devices by selecting 16 studies from ERIC, IEEE Xplore, IGI, Sage, ScienceDirect and Springer. The study *Cho et al. (2018)* finalized 20 studies to see the effect of mobile devices in language learning on students' achievement, and ERIC, EBSCOhost, JSTOR and ProQuest repositories were filtered for the literature search. This systematic literature review aims to explore the Web of Science core collection for

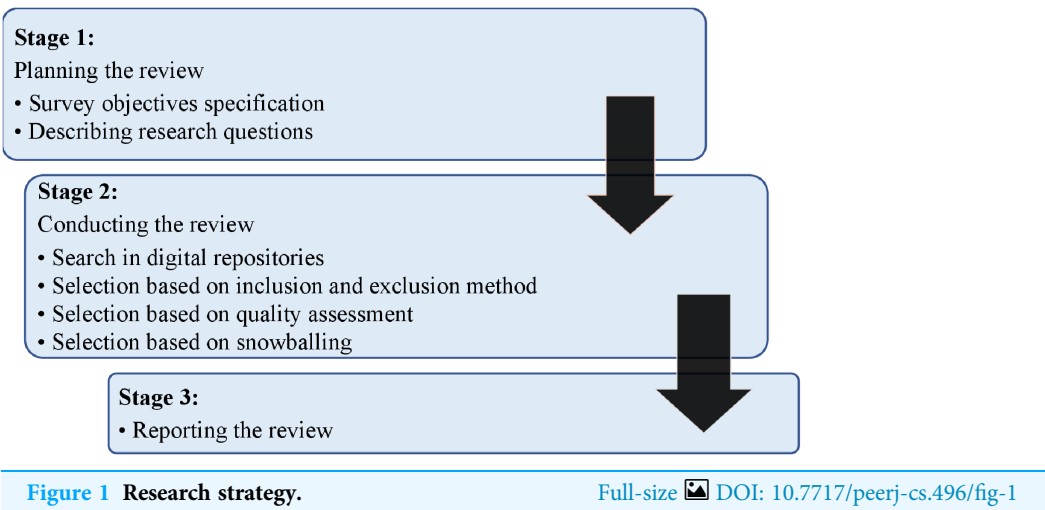

**Figure 1  Research strategy.**               

high-quality literature search to know the research trends of Mobile and Gamification based language learning. Moreover, a quality assessment of the selected articles was conducted along with the discussion of research frameworks/models and teaching and learning tools at all education levels.

# RESEARCH METHODOLOGY

This survey implemented recommendations for systematic reviews given in information engineering analysis by (*Brereton et al., 2007*). Based on these criteria, a search method was defined to eliminate possible study biases after finalizing research queries. Within this procedure, three critical phases of our analysis approach were to prepare, conduct, and review the study, as shown in Fig. 1 and discussed in the following sections.

## Review plan

An appropriate search strategy was created for all related studies. As shown in Figs. 1 and 2, the analysis methods indicate search procedures for the associated articles, describing the classification system and mapping of items. This study follows an organized approach:

- Research objectives
- Specifying research questions (RQs)
- Organizing searches of databases
- Studies selection
- Screening relevant studies
- Data extraction
- Results synthesizing
- Finalizing the review report

    **(i)**   RQ1 attempted to report our objective to develop an articles' library related to the MALL pupils and make the dataset accessible to other scholars. Furthermore, significant work was identified that provided direction to investigate students'

issues in learning English. The answer to RQ1 discovered trends, geographical areas, and publication channels in the articles.

(ii) To identify the theories and frameworks/conceptual models used for MALL research and relate them with one another and to different application areas for MALL. The solution of RQ2 provided the answer to this objective.

(iii) RQ3 attempted to identify different target application areas of MALL from teaching and learning perspectives. Furthermore, other modes of exposition were identified for these MALL applications.

(iv) How the researchers accommodated different MALL applications content focusing on reading, writing, speaking, and listening perspectives. For this purpose, RQ4 attempted to achieve this objective.

(v) RQ5 attempted to outline the standard process, tools, and instruments used to evaluate MALL applications. Furthermore, MALL applications' evaluation measures were identified concerning different perspectives, including teaching and learning, and technical perspectives.

(vi) RQ6 attempted to perform a comparative analysis to evaluate the effectiveness of simple mobile application-based language learning with gamified mobile applications for language learning.

The research objectives and motivations of this research are transformed into relevant research questions (RQ), as shown in Table 2.

## Review conduct

There were four steps in the review process formulated. In the first step, examination was made from web of science (WoS, 2020) with SCI-Expended, SCIE, ESCI and A&HCI indices consisting of high-impact research papers, for relevant primary studies. In the second step, the collection of studies was filtered based on predefined inclusion/exclusion criteria. We also established quality assessment standards to boost further the consistency in the third step of our analysis. Backward snowballing was then carried out in the final fourth step, to retrieve relevant candidate articles.

### Automated search in web of science (WoS core collection)

A systematic investigation was made to filter irrelevant research and obtain adequate information. Our source was a curated database, Web of Science Core library that included over 21,100 peer-reviewed journals, top-class academic journals distributed worldwide (including Open Access journals), covering more than 250 disciplines (WoS, 2020). WoS is a tool that helps users collect, interpret, and share information from databases promptly (Princeton University, 2020). To conduct an SLR in an organized and timely manner, the researcher used this platform to retrieve the research articles by incorporating 'AND' and 'OR' Boolean operators with keywords to develop a search string. Figure 3 presented an overview of the search result obtained from the Web of Science.

Table 3 lists the final search string incorporated 'AND' and 'OR' Boolean operators with keywords, used to explore WoS Core Collection. Only titles were searched from the

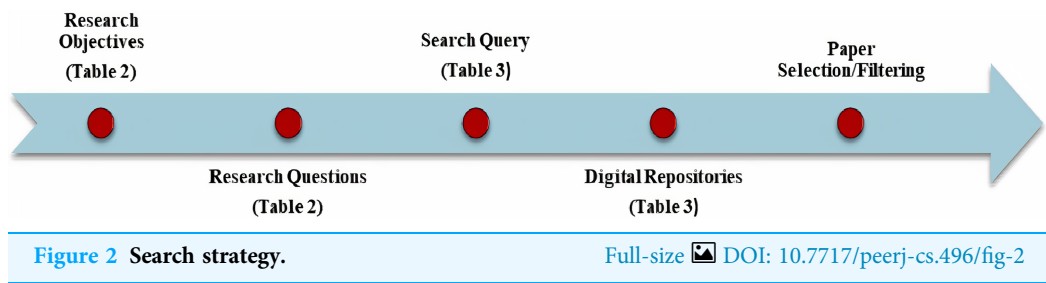

**Figure 2 Search strategy.**               

| Table 2 Research questions (RQs). | |
|---|---|
| **(RQ)  RQ statement** | **Objectives and motivation** |
| RQ1:  What were the high-quality publication channels for MALL research, and which geographical areas have been targeting MALL research over the years? | The objectives of RQ1 were to search for high-quality research articles through major publications channels for MALL research. Furthermore, the quality assessment for the selected articles and the meta-information extracted useful statistics, including the geographical areas and publication, trended over the years. |
| RQ2:  What were the widely used theories, models, and frameworks proposed or adopted for MALL research? | To identify the theories and frameworks/conceptual models used for MALL research and related them to different application areas for MALL. |
| RQ3:  What were different application domains for the MALL application, and in which various forms were these applications exposed for the end-users? | To identify different target application areas of MALL from teaching and learning perspectives. Furthermore, identify different modes of exposition for these MALL applications. |
| RQ4:  What was the specific content adopted for teaching and learning in MALL research? | To accommodate different contents for MALL applications focusing on reading, writing, speaking, and listening perspectives. |
| RQ5:  How and in what different perspectives the MALL applications were evaluated, and what were the evaluation measures and tools used for their evaluation? | Outline the standard process, tools, and instruments used for the evaluation of MALL applications. Furthermore, identify the evaluation measures for MALL applications concerning different perspectives, including teaching and learning and technical perspectives. |
| RQ6:  Compare the usage of simple mobile applications with gamified applications (Serious Game) for language learning? | To perform a comparative analysis to evaluate the effectiveness of simple mobile application-based language learning with gamified mobile applications for language learning. |

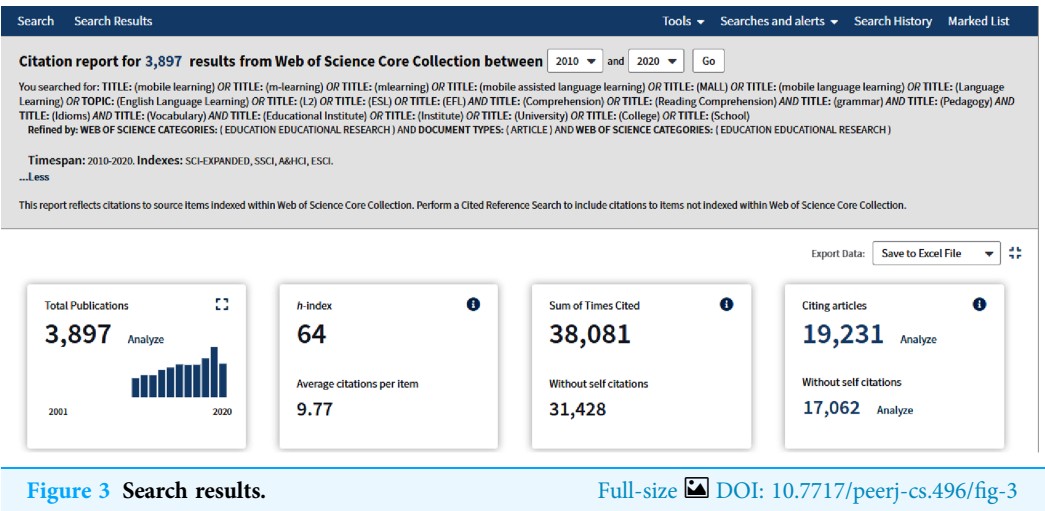

**Figure 3 Search results.**               

**Table 3 Digital library search strategy.**

| Digital library | Search query | Applied filter |
|---|---|---|
| (WoS Core Collection) SCI-Expended SCIE ESCI A&HCI | **TITLE:** (mobile learning) OR **TITLE:** (m-learning) OR **TITLE:** (m-learning) OR **TITLE:** (mobile assisted language learning) OR **TITLE:** (MALL) OR **TITLE:** (mobile language learning) OR **TITLE:** (Language Learning) OR **TITLE:** (English Language Learning) OR **TITLE:** (L2) OR **TITLE:** (ESL) OR **TITLE:** (EFL) AND **TITLE:** (Comprehension) OR **TITLE:** (Reading Comprehension) AND **TITLE:** (grammar) AND **TITLE:** (Pedagogy) AND **TITLE:** (Idioms) AND **TITLE:** (Vocabulary) AND **TITLE:** (Educational Institute) OR **TITLE:** (Institute) OR **TITLE:** (University) OR **TITLE:** (College) OR **TITLE:** (School) OR **TITLE:** (Elementary School) OR **TITLE:** (Primary School) Defining | 2010–2020 |

database, and a filter of indices and period were applied to restrict the search query for the study.

### Selection based on inclusion/exclusion criteria

#### 1. Inclusion criteria

The paper comprised in the review must be in MALL, mobile learning, and m-learning that must target the research questions. Paper published in the journals or conferences also from 2010 to 2020 was included in the review. Papers discussing MALL at school, college, and university level, focusing on learning, teaching, learning, and teaching (both), games (Example: Serious Game, Mobile games, learning application) were also included in the review.

#### 2. Exclusion criteria

The articles were excluded not written in English and did not discuss or focused MALL, mobile learning or m-learning in schools, colleges, and universities to teach and learn English. A selection process of relevant articles for inclusion/exclusion criteria in detail was shown in Fig. 4.

### Selection based on quality assessment

The collection of appropriate studies based on quality assessment (QA) was considered the critical step for carrying out any review. As the fundamental studies differed in nature, the essential assessment tool (*Fernandez, Insfran & Abrahão, 2011*) and (*Ouhbi et al., 2015*) used to conduct QA were also supplemented in our analysis by quantitative, qualitative, and mixed approaches. The accuracy of the selected records was determined using a QA questionnaire. The first author conducted the QA, using the following parameters for each study:

(a) If the analysis led to MALL, mobile language learning, m-learning, the result was indeed (1), otherwise (0).

(b) When there were simple answers for the MALL, mobile learning, m-learning of English, the analysis provided the following scores: 'Yes (2),' 'Limited (1)' and 'No (0).'

(c) If the studies provided an empirical result, then award (1) else score (0).

(d) Studies analyzed concerning graded rankings of countries and journals (*SCImago Journal & Country Rank (SJR), 2018*) and conferences in computer science (*CORE*

*Conference Portal, 2018*). Table 4 indicated potential findings for publications from known and reliable sources.

After combining the scores to the above questions, a final score (between 0 and 8) was determined for each study. Only papers included with four or more ratings in the final results.

### Selection based on snowballing

Following the standard assessment, backward snowballing was performed, employing a reference list from any completed analysis to retrieve papers (*Mehmood et al., 2020*) and chose only those significant articles that met inclusion/exclusion requirements. After reading the introduction and then other portions of the document, the article's inclusion/ exclusion was determined.

## Review report

This section provided an overview of the selected studies.

### Overview of intermediate selection process outcome

MALL was a very active topic, and our analysis approach had to extract relevant research empirically and systematically from the Web of Science core collection. The next step of our systematic analysis was compiling records that form the foundation for this analysis. Approximately 57,000 papers were examined from the archive by providing the keywords for 2010– 2020. After creating a knowledge base from the digital library (Web of Science), the author reviewed the title, abstract, and accompanying complete document for each search result, as needed. During this process, irrelevant papers or papers of less than four pages were eliminated. During the inspection process, selected documents in the fields of MALL, mobile learning, and m-learning were read extensively to assess their significance & contribution and then created a comprehensive knowledge foundation of papers based on their findings to accomplish this research's core objective.

### Overview of selected studies

Table 5 presented significant results of the primary search, filtering, and reviewed processes that included Web of Science indices. At the filtering/inspection stage, this amount decreased to 63 articles by the automatic search.

## ASSESSMENT AND DISCUSSION OF RESEARCH QUESTIONS

In this section, finalized 67 primary research studies based on our research questions were scrutinized.

## RQ1: What are the high-quality publication channels for MALL research, and which geographical areas have been targeting MALL research over the years?

The analysis of MALL with the integration of game elements in learning tools, methods, content, and the theoretical perspective choice was a crucial challenge for scholars to use in education. Identifying fine publication sites and scientometric analysis based on meta-

**Figure 4** Selection of relevant articles using Systematic Review Process.

**Table 4 Rating for stable and recognized publication sources.**

| Item. no. | Publication source | +4 | +3 | +2 | +1 | +0 |
|---|---|---|---|---|---|---|
| 1 | Journals | Q1 | Q2 | Q3 | Q4 | No JCR Ranking |
| 2 | Conferences | Core A | Core A | Core B | Core C | Not in Core Ranking |

information in MALL domain was required for the purpose. This section consisted of perceptive knowledge of research publication sources, types, years, grade level distribution, geographical distribution, publication channel-wise distribution of selected studies to analyze MALL research.

The studies finalized from the Web of Science (core collection) were presented yearly, as shown in Table 6 and Fig. 5. Twelve was the maximum number of publications selected from the year 2019, out of the total 67, indicated more interest in developing MALL with integrating games in teaching and learning. However, less interest in MALL with game research integration was observed in 2010–2016 and 2018–2020, resulting in less improvement in teaching and learning in relevance to students and market needs.

**Table 5 Selection phases and results.**

| Phase | Selection | Selection criteria | Indexes: SCI-EXPANDED, SSCI, A&HCI, ESCI |
|---|---|---|---|
| 1 | Search | Keywords (Figure) | 57,364 |
| 2 | Filtering | Title | 3,897 |
| 3 | Filtering | Abstract | 275 |
| 4 | Filtering | Introduction and Conclusion | 167 |
| 5 | Inspection | Full Article | 67 |

**Table 6 Identified publications by year.**

| Year | 2010 | 2011 | 2012 | 2013 | 2014 | 2015 | 2016 | 2017 | 2018 | 2019 | 2020 | Total |
|---|---|---|---|---|---|---|---|---|---|---|---|---|
| Number of publication | 2 | 3 | 6 | 6 | 6 | 2 | 6 | 7 | 10 | 12 | 7 | 67 |

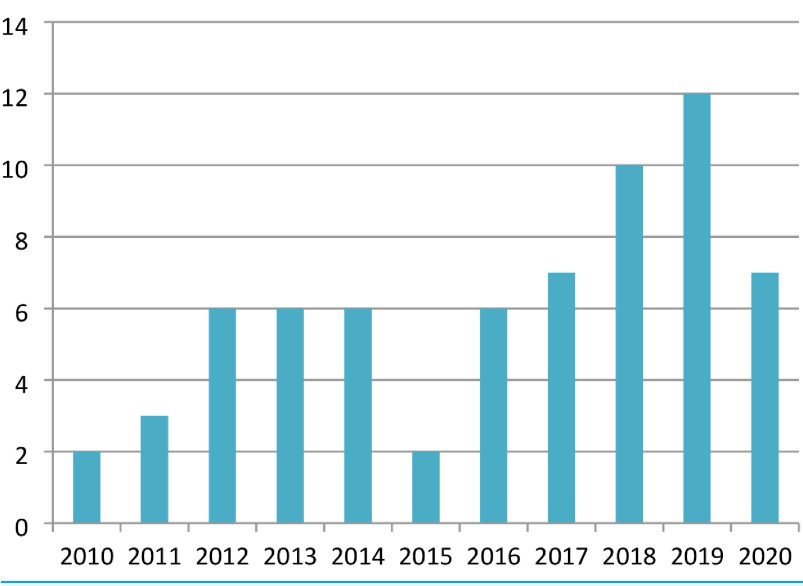

**Figure 5 Graph of identified studies by year.**

Table 7 and Fig. 6 present the geographically distributed studies. The majority of publications, or 39 out of 67, were from different Asian countries, whereas European countries published 15 reviews. North American countries published eight studies, while Africa and Ocean-continent have three and two studies published, respectively.

The data presented in Table 8 showing that the maximum number of articles were from highly recognized journals indexed in the Web of Science. Only one item was from a good ranked conference. Computer & Education journal was on the top of the list from which seven papers were selected and next, the Educational Technology & Society journal with four articles.

According to conditions defined in "Research Methodology" B.3, each finalized study's QA score was granted, as shown in Table 4, offering the QA score ranging from 4–8,

| Item no. | Sub-continent | Countries | Number of publication |
|---|---|---|---|
| 1 | Asia | Taiwan | 14 |
| | | Turkey | 5 |
| | | Malaysia | 4 |
| | | Hong Kong | 3 |
| | | Singapore | 3 |
| | | China | 2 |
| | | Saudi Arab | 3 |
| | | Pakistan | 1 |
| | | Japan | 1 |
| | | Iran | 1 |
| | | India | 1 |
| | | Israel | 1 |
| 2 | Europe | Netherland | 3 |
| | | UK | 3 |
| | | Czech Republic | 1 |
| | | France | 1 |
| | | Luxembourg | 1 |
| | | Spain | 1 |
| | | Belgium | 1 |
| | | Norway | 1 |
| | | Germany | 1 |
| | | Estonia | 2 |
| 3 | Africa | Algeria | 1 |
| | | South Africa | 1 |
| | | Morocco | 1 |
| 4 | North America | US | 7 |
| | | Canada | 1 |
| 5 | Ocean-continent | New Zealand | 1 |
| | | Australia | 1 |
| **Total** | | | **67** |

Table 7 Publications by geographic areas.

with less than four discarded scores. MALL researcher might find this QA supportive to choose related studies while addressing its usage and challenges. Articles published in Q1 journals mostly scored maximum while scoring four from less recognized journals but relevant to the subject matter. A total of 26 out of 67 scored maximum (i.e., eight (8), indicating fulfillment of all QA criteria, whereas nine (9) studies scored four out of 67, which is the lowest in QA).

Table 9 presented the overall classification output and QA of finalized studies, and Table 10 showed the overall quality assessment score. Studies were classified based on five factors: the empirical type/method, research type, and method. Categories of research types were; Evaluation framework, Evaluation research, Solution proposal, and Review.

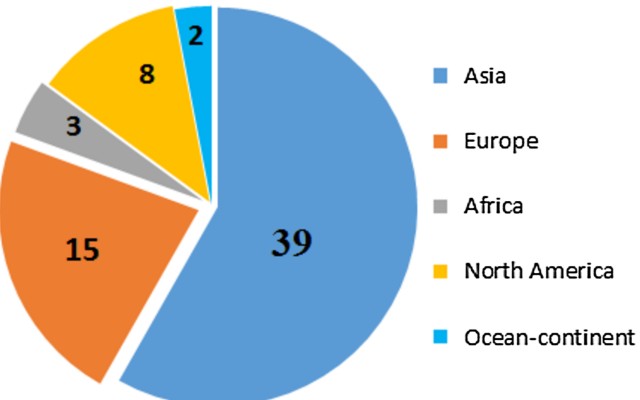

Figure 6 Graph of identified studies by continent.

The taxonomy presented in section V was constructed on these defined research types. Studies that analytically validated their results from the statistical analysis, experimentations, surveys, or case studies to increase their quality standards awarded a score. In category (c) of quality assessment criteria, only 8 out of 67 reviews did not present empirical results, thus awarded zero scores. Only five studies scored zero for category (d) of quality assessment criteria; the rest of them scored higher, indicating competent sources. Four (4) recorded the lowest score awarded for a study.

## RQ2. What are the widely used theories, models, and frameworks proposed or adopted for MALL research?

This section provided the framework/conceptual model based on the situation (proposed/adopted by the researcher) after an essential discussion of the theories, frameworks, and models.

### THEORIES

*Theory of planned behavior*: The Theory of planned behavior (TPB) noted that behavioral intentions motivated human behavior. Behavioral intentions rely on three determinants; an individual's mood, cultural norms, and perceived influence (*Cheon et al., 2012*).

*Sociocultural theory*: Sociocultural theory recognized human evolution as socially influenced by cooperation with more professional members of society, through which children learned their cultural norms, ideologies, and problem-solving techniques (*Kearney et al., 2012*) (*Mcleod, 2020*).

*Experiential Learning Theory*: The concept of experiential learning was implemented in a learning game. Players performed exercises to seek information within a gameplay environment that made the learning process enjoyable, engaging, and interesting (*Tsai et al., 2016*).

*Dual-Coding theory*: The dual-coding theory suggested that the verbal and imagery systems (mental images and representations) could be co-activated as rich and substantive referential relations connected dual-coded objects (*Teng, 2018*).

**Table 8  Publication sources.**

| Item no. | Publication source | Channel | No. of articles |
|---|---|---|---|
| 1 | Computers & Education | Journal | 7 |
| 2 | Educational Technology & Society | Journal | 4 |
| 3 | Computer-Assisted Language Learning | Journal | 3 |
| 4 | Recall | Journal | 3 |
| 5 | Education and Information Technologies | Journal | 3 |
| 6 | Journal of Asia TEFL | Journal | 3 |
| 7 | International Journal of Advanced Computer Science and Applications | Journal | 3 |
| 8 | Journal of Computer Assisted Learning | Journal | 2 |
| 9 | Interactive Learning Environments | Journal | 2 |
| 10 | CALICO Journal | Journal | 2 |
| 11 | Technology Pedagogy and Education | Journal | 2 |
| 12 | International Review of Research in Open and Distributed Learning | Journal | 2 |
| 13 | International Journal of Mobile and Blended Learning | Journal | 2 |
| 14 | Education Sciences | Journal | 2 |
| 15 | Journal of Educational Computing Research | Journal | 2 |
| 16 | Language Learning Journal | Journal | 2 |
| 17 | Research in Learning Technology | Journal | 1 |
| 18 | International Journal of Game-Based Learning | Journal | 1 |
| 19 | Turkish Online Journal of Educational Technology | Journal | 1 |
| 20 | British Journal of Educational Technology | Journal | 1 |
| 21 | Interactive Technology and Smart Education | Journal | 1 |
| 22 | Language Culture and Curriculum | Journal | 1 |
| 23 | International Journal of Computer-Assisted Language Learning and Teaching | Journal | 1 |
| 24 | Australian Educational Researcher | Journal | 1 |
| 25 | Journal of Information Technology Education-Research | Journal | 1 |
| 26 | International Journal of Emerging Technologies in Learning | Journal | 1 |
| 27 | International Journal of Distance Education Technologies | Journal | 1 |
| 28 | Journal of Computers in Education | Journal | 1 |
| 29 | International Journal of Bilingual Education and Bilingualism | Journal | 1 |
| 30 | International Journal of Instruction | Journal | 1 |
| 31 | Journal of Language and Education | Journal | 1 |
| 32 | MIER-Journal of Educational Studies Trends and Practices | Journal | 1 |
| 33 | International Journal of Information and Communication Technology Education | Journal | 1 |
| 34 | International Journal of Continuing Engineering Education and Life-Long Learning | Journal | 1 |
| 35 | 2019 International Conference on Innovative Computing (ICIC) | Conference | 1 |
| 36 | 2019 International Conference on Computer and Information Sciences (ICCIS) | Conference | 1 |
| 37 | 2019 12th International Conference on Information & Communication Technology and System (ICTS) | Conference | 1 |
| 38 | 2018 International Conference on Current Trends towards Converging Technologies (ICCTCT) | Conference | 1 |
| 39 | 2020 IEEE 20th International Conference on Advanced Learning Technologies (ICALT) | Conference | 1 |

**Table 9 Quality assessment.**

| Ref. | Classification | | | | | Quality assessment | | | | |
|---|---|---|---|---|---|---|---|---|---|---|
| | P. channel | Publication year | Research type | Empirical type/method | Methodology | (a) | (b) | (c) | (d) | Score |
| (*Cheon et al., 2012*) | Research Journal | 2012 | Evaluation Framework | No | Formulation of conceptual model | 1 | 0 | 0 | 4 | 5 |
| (*Kearney et al., 2012*) | Research Journal | 2012 | Evaluation Framework | No | Formulation of Pedagogical Framework | 1 | 0 | 0 | 3 | 4 |
| (*Martin & Ertzberger, 2013*) | Research Journal | 2013 | Evaluation Research | Survey | Statistical Analysis | 1 | 2 | 1 | 4 | 8 |
| (*Liu, Li & Carlsson, 2010*) | Research Journal | 2010 | Evaluation Research | Survey | Questionnaire | 1 | 2 | 1 | 4 | 8 |
| (*Sandberg, Maris & Geus, 2011*) | Research Journal | 2011 | Evaluation Research | Survey | Questionnaire | 1 | 2 | 1 | 4 | 8 |
| (*Hsu, Hwang & Chang, 2013*) | Research Journal | 2013 | Solution Proposal | Experiment | Personalized recommendation-based approach | 1 | 2 | 1 | 4 | 8 |
| (*Wong & Looi, 2010*) | Research Journal | 2010 | Evaluation Research | Survey | Questionnaire | 1 | 2 | 1 | 4 | 8 |
| (*Holden & Sykes, 2011*) | Research Journal | 2011 | Evaluation Research | Survey | Interview and Observation | 1 | 1 | 1 | 3 | 6 |
| (*Zhang, Song & Burston, 2011*) | Research Journal | 2011 | Evaluation Research | Experiment | Statistical Analysis | 1 | 1 | 1 | 1 | 4 |
| (*Sandberg, Maris & Hoogendoorn, 2014*) | Research Journal | 2014 | Solution Proposal | Experiment | Game | 1 | 2 | 1 | 4 | 8 |
| (*Huang et al., 2016*) | Research Journal | 2016 | Evaluation Research | Experiment + Survey | Learning tool and Questionnaire | 1 | 2 | 1 | 4 | 8 |
| (*Lin, 2014*) | Research Journal | 2014 | Solution Proposal | Experiment + Survey | Reading tool and Questionnaire | 1 | 2 | 1 | 4 | 8 |
| (*Huang et al., 2011*) | Research Journal | 2012 | Evaluation Research | Survey | Questionnaire | 1 | 2 | 1 | 4 | 8 |
| (*Dashtestani, 2015*) | Research Journal | 2016 | Evaluation Research | Survey | Questionnaire, Semi-Structured Interview and Observation | 1 | 2 | 1 | 4 | 8 |
| (*Liu & Chen, 2014*) | Research Journal | 2015 | Evaluation Research | Experiment and Survey | Photo taking | 1 | 1 | 1 | 4 | 7 |
| (*Liakin, Cardoso & Liakina, 2014*) | Research Journal | 2015 | Solution Proposal | Experiment | Instructional tool | 1 | 1 | 1 | 4 | 7 |
| (*Dennen & Hao, 2014*) | Research Journal | 2014 | Solution Proposal | Framework 0 | Framework proposed | 1 | 0 | 0 | 4 | 5 |
| (*Wong, 2013*) | Research Journal | 2013 | Evaluation Research | Survey | Questionnaire | 1 | 2 | 1 | 4 | 8 |
| (*Bohm & Constantine, 2016*) | Research Journal | 2016 | Solution Proposal | Model 0 | Model Proposed | 1 | 0 | 0 | 3 | 4 |
| (*Wu, 2018*) | Research Journal | 2018 | Solution Proposal | Model | Game (Vocabulary) | 1 | 2 | 0 | 4 | 7 |
| (*Huang, 2014*) | Research Journal | 2014 | Evaluation Research | Survey | Questionnaire | 1 | 2 | 1 | 4 | 8 |
| (*Lai & Zheng, 2017*) | Research Journal | 2018 | Evaluation Research | Survey | Online Survey and Interview | 1 | 2 | 1 | 4 | 8 |

(Continued)

| Ref. | Classification | | | | | Quality assessment | | | | |
|------|----------------|--|--|--|--|--------------------|--|--|--|--|
| | P. channel | Publication year | Research type | Empirical type/method | Methodology | (a) | (b) | (c) | (d) | Score |
| (Avci & Adiguzel, 2017) | Research Journal | 2017 | Evaluation Research | Survey | Interview and Focus group discussion | 1 | 2 | 1 | 0 | 4 |
| (Chang et al., 2013) | Research Journal | 2013 | Solution Proposal | Experiment | Model Used | 1 | 2 | 1 | 4 | 8 |
| (Elgün-Gündüz, Akcan & Bayyurt, 2012) | Research Journal | 2012 | Evaluation Research | Statistical Analysis | Assessment Test | 1 | 2 | 1 | 3 | 7 |
| (Amer, 2014) | Research Journal | 2014 | Solution Proposal | Experiment + Survey | Game, Questionnaire and Interview | 1 | 2 | 1 | 4 | 8 |
| (Petersen, Procter-Legg & Cacchione, 2013) | Research Journal | 2013 | Solution Proposal | No | Informal | 1 | 1 | 0 | 2 | 4 |
| (Tsai et al., 2016) | Research Journal | 2017 | Solution Proposal | Experiment | Game | 1 | 2 | 1 | 4 | 8 |
| (Kim, Ruecker & Kim, 2017) | Research Journal | 2017 | Evaluation Research | Survey | Questionnaire | 1 | 2 | 1 | 2 | 6 |
| (Wang, Zou & Xing, 2014) | Research Journal | 2014 | Evaluation Research | Survey | Interviews | 1 | 2 | 1 | 2 | 6 |
| (Teng, 2018) | Research Journal | 2019 | Solution Proposal | Experiment | Captioned Video | 1 | 2 | 1 | 4 | 8 |
| (Botero, Questier & Zhu, 2018) | Research Journal | 2019 | Evaluation Research | Survey | Questionnaire and Interviews | 1 | 2 | 1 | 4 | 8 |
| (Yurdagül & Öz, 2018) | Research Journal | 2018 | Evaluation Research | Survey | Questionnaire | 1 | 2 | 1 | 2 | 6 |
| (Gafni, Achituv & Rahmani, 2017) | Research Journal | 2017 | Evaluation Research | Survey | Questionnaire | 1 | 2 | 1 | 3 | 7 |
| (Zhang, 2016) | Research Journal | 2016 | Evaluation Research | Survey | Questionnaire | 1 | 2 | 1 | 2 | 6 |
| (Uematsu, 2012) | Research Journal | 2012 | Evaluation Research | Survey | Test and Interviews | 1 | 2 | 1 | 2 | 6 |
| (Pham, Nguyen & Chen, 2017) | Research Journal | 2018 | Evaluation Research | No | Review | 1 | 1 | 0 | 4 | 6 |
| (Çakmak & Erçetin, 2017) | Research Journal | 2018 | Evaluation Research | Experiment + Survey | Statistical Analysis | 1 | 2 | 1 | 4 | 8 |
| (Shih, 2017) | Research Journal | 2017 | Evaluation Research | Survey | Questionnaire and Class Observation | 1 | 2 | 1 | 2 | 6 |
| (Ou-Yang & Wu, 2016) | Research Journal | 2017 | Evaluation Research | Survey + Experiment | Questionnaire and Log Analyzer | 1 | 2 | 1 | 4 | 8 |
| (Quan, 2016) | Research Journal | 2016 | Solution Proposal | Experiment + Survey | Tool for academic test and Questionnaire with Interviews | 1 | 2 | 1 | 0 | 4 |
| (Tragant et al., 2015) | Research Journal | 2016 | Review | No | Informal | 1 | 1 | 0 | 4 | 6 |
| (Chen & Lin, 2018) | Research Journal | 2018 | Evaluation Research | Survey | Questionnaire | 1 | 2 | 1 | 2 | 6 |
| (Bouchaib, Ahmadou & Abdelkader, 2018) | Research Journal | 2018 | Evaluation Research | Survey | Questionnaire | 1 | 2 | 1 | 3 | 7 |

| Ref. | Classification | | | | | Quality assessment | | | | |
|------|------|------|------|------|------|------|------|------|------|------|
| | P. channel | Publication year | Research type | Empirical type/method | Methodology | (a) | (b) | (c) | (d) | Score |
| (Hazaea & Alzubi, 2018) | Research Journal | 2018 | Evaluation Research | Survey | Interview | 1 | 2 | 1 | 1 | 5 |
| (Kohnke, Zhang & Zou, 2019) | Research Journal | 2019 | Evaluation Research | Survey | Questionnaire | 1 | 2 | 1 | 2 | 6 |
| (Chen, Liu & Huang, 2019) | Research Journal | 2019 | Evaluation Research | Survey | Questionnaire and Interviews | 1 | 2 | 1 | 4 | 8 |
| (Kirsch & Izuel, 2017) | Research Journal | 2019 | Evaluation Research | Experiment | Formal | 1 | 2 | 1 | 3 | 7 |
| (Kalogirou, Beauchamp & Whyte, 2017) | Research Journal | 2019 | Evaluation Research | Experiment | Formal | 1 | 2 | 1 | 3 | 7 |
| (Makoe & Shandu, 2018) | Research Journal | 2018 | Evaluation Research | Survey | Interview | 1 | 2 | 1 | 0 | 4 |
| (Shahbaz & Khan, 2017) | Research Journal | 2017 | Evaluation Research | Experiment | Statistical Analysis | 1 | 2 | 1 | 0 | 4 |
| (Fisser, Voogt & Bom, 2012) | Research Journal | 2013 | Solution Proposal | Experiment | Game | 1 | 2 | 1 | 4 | 8 |
| (Ng et al., 2020) | Research Journal | 2020 | Evaluation Research | Experiment | Statistical Analysis | 1 | 2 | 1 | 4 | 8 |
| (Bourekkache & Kazar, 2020) | Research Journal | 2020 | Solution Proposal | Experiment | Learning tool for English | 1 | 1 | 1 | 2 | 5 |
| (Klimova & Polakova, 2020) | Research Journal | 2020 | Evaluation Research | Survey | Questionnaire | 1 | 2 | 1 | 2 | 6 |
| (Önal, Çevik & Şenol, 2019) | Research Journal | 2019 | Solution Proposal | Experiment | Game (Tenses) | 1 | 2 | 1 | 4 | 8 |
| (Zhang & Pérez-Paredes, 2019) | Research Journal | 2019 | Evaluation Research | Survey | Questionnaire and Interviews | 1 | 2 | 1 | 4 | 8 |
| (Chu, Wang & Wang, 2019) | Research Journal | 2019 | Evaluation Research | Survey | Questionnaire | 1 | 2 | 1 | 4 | 8 |
| (Ramadoss & Wang, 2012) | Research Journal | 2012 | Evaluation Research | Survey | Questionnaire, Interviews and Log files | 1 | 2 | 1 | 1 | 5 |
| (Ishaq et al., 2020a) | Research Journal | 2020 | Evaluation Research | Survey | Questionnaire | 1 | 2 | 1 | 1 | 5 |
| (Ishaq et al., 2020b) | Research Journal | 2020 | Evaluation Research | Survey | Questionnaire and Interviews | 1 | 2 | 1 | 1 | 5 |
| (Ishaq et al., 2019) | Conference | 2019 | Evaluation Research | Survey | Questionnaire | 1 | 2 | 1 | 0 | 4 |
| (Ishaq et al., 2020d) | Research Journal | 2020 | Evaluation Research | Survey | Questionnaire and Interviews | 1 | 2 | 1 | 1 | 5 |
| (Yanes & Bououd, 2019) | Conference | 2019 | Evaluation Research | Experiment | SWOT Analysis | 1 | 2 | 1 | 1 | 5 |
| (Angelia & Suharjito, 2019) | Conference | 2019 | Evaluation Research | Survey | Questionnaire | 1 | 1 | 1 | 2 | 5 |
| (Tshering et al., 2018) | Conference | 2018 | Evaluation Research | Experiment | Interview | 1 | 2 | 1 | 2 | 6 |
| (Krasulia & Saks, 2020) | Conference | 2020 | Evaluation Research | Survey | Questionnaire | 1 | 1 | 1 | 3 | 6 |

**Table 10 Quality assessment score.**

| References | Score | Total |
|---|---|---|
| (*Martin & Ertzberger, 2013*) (*Liu, Li & Carlsson, 2010*) (*Sandberg, Maris & Geus, 2011*) (*Hsu, Hwang & Chang, 2013*) (*Wong & Looi, 2010*) (*Sandberg, Maris & Hoogendoorn, 2014*) (*Huang et al., 2016*) (*Lin, 2014*) (*Huang et al., 2011*) (*Dashtestani, 2015*) (*Wong, 2013*) (*Huang, 2014*) (*Lai & Zheng, 2017*) (*Chang et al., 2013*) (*Amer, 2014*) (*Tsai et al., 2016*) (*Teng, 2018*) (*Botero, Questier & Zhu, 2018*) (*Çakmak & Erçetin, 2017*) (*Ou-Yang & Wu, 2016*) (*Chen, Liu & Huang, 2019*) (*Fisser, Voogt & Bom, 2012*) (*Ng et al., 2020*) (*Önal, Çevik & Şenol, 2019*) (*Zhang & Pérez-Paredes, 2019*) (*Chu, Wang & Wang, 2019*) | 8 | 26 |
| (*Liu & Chen, 2014*) (*Liakin, Cardoso & Liakina, 2014*) (*Wu, 2018*) (*Elgün-Gündüz, Akcan & Bayyurt, 2012*) (*Gafni, Achituv & Rahmani, 2017*) (*Bouchaib, Ahmadou & Abdelkader, 2018*) (*Kirsch & Izuel, 2017*) (*Kalogirou, Beauchamp & Whyte, 2017*) | 7 | 8 |
| (*Holden & Sykes, 2011*) (*Kim, Ruecker & Kim, 2017*) (*Wang, Zou & Xing, 2014*) (*Yurdagül & Öz, 2018*) (*Zhang, 2016*) (*Uematsu, 2012*) (*Pham, Nguyen & Chen, 2017*) (*Shih, 2017*) (*Tragant et al., 2015*) (*Chen & Lin, 2018*) (*Kohnke, Zhang & Zou, 2019*) (*Klimova & Polakova, 2020*) (*Tshering et al., 2018*) (*Krasulia & Saks, 2020*) | 6 | 14 |
| (*Cheon et al., 2012*) (*Dennen & Hao, 2014*) (*Hazaea & Alzubi, 2018*) (*Bourekkache & Kazar, 2020*) (*Ramadoss & Wang, 2012*) (*Ishaq et al., 2020a*) (*Ishaq et al., 2020b*) (*Ishaq et al., 2020d*) (*Yanes & Bououd, 2019*) (*Angelia & Suharjito, 2019*) | 5 | 10 |
| (*Kearney et al., 2012*) (*Zhang, Song & Burston, 2011*) (*Bohm & Constantine, 2016*) (*Avci & Adiguzel, 2017*) (*Petersen, Procter-Legg & Cacchione, 2013*) (*Quan, 2016*) (*Makoe & Shandu, 2018*) (*Shahbaz & Khan, 2017*) (*Ishaq et al., 2019*) | 4 | 9 |

*Theory of cognitive style:* Two types of learning, field independence and field dependence, were based on critical and interpersonal personality characteristics. Field-independent students were independent-minded, and the social environment could not readily affect their learning style. Field-dependent students tended to learn in visual settings or scenarios (*Ou-Yang & Wu, 2016*).

*Attribution theory:* Bouchaib (*Bouchaib, Ahmadou & Abdelkader, 2018*) used Weiner attribution theory (1992) as a framework that provided failure or success reasons or explanations for people in education. Moreover, it was also a social cognitive theory of motivation.

*Instructional theory:* The instructional theory sought to explain how people can learn, evolve, and established environments that promote learning opportunities and strengthen teaching (*Fisser, Voogt & Bom, 2012*) (*Top Hat, 2019*).

*Frameworks & models*

*'Here and now' mobile learning framework:* Three characteristics (Engaging, Authentic, and Informal) Framework identified the effect of mobile learning on the learning environment (*Martin & Ertzberger, 2013*).

*Mobile English learning outcome (MELO) framework:* This research developed the Mobile English Learning Objective (MELO) to examine the correlation between learners' understanding of mobile technologies using playfulness, resistance to change, and self-management constructs of the Framework (*Huang et al., 2011*).

*M-COPE:* It is a study that came up with the M-COPE framework to support teachers in building mobile learning experiences using the ADDIE model, which allowed teachers to understand five main aspects of mobile learning: Mobile affordance, Ethics, Pedagogy, and Outcomes (*Dennen & Hao, 2014*).

*Mobile English learning continuance intention (MELCI):* The study came up with TAM components (*Davis, 1989*) adopted in mobile English learning. Continuance intention

framework to examine the influence of mobile learning satisfaction with self-management of learning (*Huang, 2014*).

*Pedagogical Framework:* Task-based language teaching (TBLT) was a pedagogical framework concerning language instruction strategy that focused on curricular design units and an instructional cycle aspect. It focused on the interaction method, which implied that the negotiation of importance led to a detailed and internationally changed input (*Chen & Lin, 2018*).

*The affordance framework:* The affordance framework, which comprised pedagogical, social, and technological components, examined the ICT tools affordance for language learning (*Ramadoss & Wang, 2012*).

*Technology Acceptance Model (TAM):* The TAM demonstrated many knowledge predictors by many researchers, focusing on the rational action theory. TAM has five components: perceived ease of usage, perceived effectiveness, the intention of behavior, intention to use, and discrete use. The model played an essential role in forecasting the use of various factors (*Chang et al., 2013*).

*Extended Technology Acceptance Model:* The study examined mobile applications' impact on students' usage intention, perceived ease of use (PEOU), and usefulness. The research outcome was an extended technology acceptance model (*Bohm & Constantine, 2016*).

*ARCS Model:* A motivational model ARCS (Attention, Relevance, Confidence, and Satisfaction) that motivated learners and procedural assistance in compliance to demand used in this research. This model has three features: (1) emphasized motivation and emotional stimulation, (2) integrated with other theories and design, (3) enhanced the instruction effects and learning process (*Wu, 2018*).

*A Theoretical Model:* Discussion of two frameworks: (1) Framework for the rational analysis of mobile education (FRAME). (2) TAM and TAM 2, the author proposed a new theoretical model that focused on the technology enhancement, non-formal learning setting, and learner-centered to develop a better understanding for EFL learners using mobile English learning resources (MELR) for Chinese postgraduate students (*Zhang & Pérez-Paredes, 2019*).

In this section, the Theory, Framework, and Model used in the studies were described and summarized in Table 11.

A conceptual model based on the Theory of Planned Behavior (TPB) explained how young adults' acceptance and attitude affect their intention to embrace mobile devices in their classroom practice (*Cheon et al., 2012*). The Framework in (*Kearney et al., 2012*) criticized pedagogy, allowed a comparison of mobile approaches and pedagogical methods, and their importance to the sociocultural nature of learning in mobile learning environments validated by m-learning researchers' discussion among designers of the Framework, implementation in a project, and pedagogical experts. Similarly, three characteristics (Engaging, Authentic, and Informal) Framework were created to see mobile learning's effect on the learning environment by measuring students' achievement and attitude (*Martin & Ertzberger, 2013*). A hypothesized m-learning model was proposed based on the Technology Acceptance Model (TAM) components—Perceived Usefulness

**Table 11 Theory, framework and model used in the studies.**

| Ref. | Year | Application | Theory | Model/framework | Description |
|------|------|-------------|--------|-----------------|-------------|
| (Cheon et al., 2012) | 2012 | Learning readiness | Theory of Planned Behavior (TPB) | Conceptual model, based on the Theory of Planned Behavior (TPB) | A conceptual model based on the Theory of Planned Behavior (TPB) explained how young adults' behavior affects their intention. |
| (Kearney et al., 2012) | 2012 | Pedagogy | Iterative revision by stakeholders | Pedagogical Framework | The Framework used to criticize pedagogy, allowing the comparison of mobile practices and pedagogical methods. |
| (Martin & Ertzberger, 2013) | 2013 | Achievement and Attitude | Dual-coding Theory | 'Here and now' mobile learning framework | A three characteristics (Engaging, Authentic, and Informal) Framework was create to study mobile learning's effect on the learning environment. |
| (Liu, Li & Carlsson, 2010) | 2010 | Usefulness and Personal Innovativeness | Theory of reasoned action | A hypothesized model of m-learning adoption | A hypothesized model of m-learning is proposed based on the Technology Acceptance Model (TAM). |
| (Sandberg, Maris & Geus, 2011) | 2011 | Learning | Learning Theory | Pedagogical Framework | A pedagogical framework is adopted and used for mobile English learning for fifth-grade students. |
| (Hsu, Hwang & Chang, 2013) | 2013 | Reading | Not Available | TAM | A personalized recommendation based mobile language learning approach was proposed. |
| (Wong & Looi, 2010) | 2010 | Vocabulary | Not Available | MALL framework | Two case studies were presented based on MALL framework. |
| (Holden & Sykes, 2011) | 2011 | Spanish language learning | Not Available | Not Available | A mobile game for learning Spanish was developed. |
| (Zhang, Song & Burston, 2011) | 2011 | Vocabulary | Learners' active role | Channell's conceptual Framework | Reexamined the effectiveness of vocabulary learning using mobile phones, Channell's Conceptual Framework was used. |
| (Sandberg, Maris & Hoogendoorn, 2014) | 2014 | Vocabulary | Not Available | Tutoring model | Mobile English learning application developed and used for vocabulary learning. |
| (Huang et al., 2016) | 2016 | Vocabulary | Motivation theory | ARCS model | A strategy for vocabulary learning on the basis of the ARCS model and motivation theory was developed. |
| (Lin, 2014) | 2014 | Reading | Multimedia learning theory | TAM | The purpose was to investigate the effect of using mobile tablet PCs for adolescent English learners. |
| (Huang et al., 2011) | 2012 | English language learning | Self-determination theory | Mobile English learning outcome (MELO) framework | The M-COPE framework allowed teachers to understand five main aspects of mobile learning: Mobile affordance, Ethics, Pedagogy, and Outcomes. |
| (Dashtestani, 2015) | 2016 | English language learning | Not Available | Not Available | The aim was to see mobile devices for English as a foreign language among Iranian students. |
| (Liu & Chen, 2014) | 2015 | English language learning | Sociocultural theory | Not Available | The study investigated the impact of photo-taking through mobiles for English phrase learning. |
| (Liakin, Cardoso & Liakina, 2014) | 2015 | Pronunciation of French | Not Available | Not Available | The study investigated the acquisition of French vowels using a mobile learning environment. |
| (Dennen & Hao, 2014) | 2014 | Pedagogy | Instructional design theory | M-COPE | The M-COPE framework allowed teachers to understand five main aspects of mobile learning: Mobile affordance, Ethics, Pedagogy, and Outcomes. |

| Ref. | Year | Application | Theory | Model/framework | Description |
|---|---|---|---|---|---|
| (*Wong, 2013*) | 2013 | Idioms | Not Available | Chans' Framework | The study presented a MALL design that highlighted students' habits and skills for making meanings from routine event pictures. |
| (*Bohm & Constantine, 2016*) | 2016 | English language learning | Not Available | Extended Technology Acceptance Model developed | The concept of the computational model and their proposed connections resulting from the use of *Davis's (1989)* TAM was introduced in this article. |
| (*Wu, 2018*) | 2018 | Vocabulary | Motivation and combined theory | ARCS Model | A motivational model, ARCS (Attention, Relevance, Confidence, and Satisfaction), that considered motivation to learners and procedural assistance, was used in this research. |
| (*Huang, 2014*) | 2014 | English language learning | Self-determination theory | Mobile English learning continuance intention (MELCI) Framework | MELCI framework was used in which components of TAM (*Davis, 1989*) were adopted to see the influence of mobile English learning satisfaction with self-management of learning. |
| (*Lai & Zheng, 2017*) | 2018 | English language learning | Not Available | Kearneys' model | The purpose of the study was to explore the use of mobile learning experiences across space and time. |
| (*Avci & Adiguzel, 2017*) | 2017 | English language learning | Not Available | Mobile-Blended Collaborative Learning model | An instant messaging application 'Whatsapp' was used to explore the language proficiency. |
| (*Chang et al., 2013*) | 2013 | English language learning | Theory of reasoned action | TAM | This study adopted the TAM by incorporating perceived convenience and curiosity factor for mobile learning. |
| (*Elgün-Gündüz, Akcan & Bayyurt, 2012*) | 2012 | Vocabulary, Grammar | Not Available | Not Available | The study explored the effect of integrated form-focused instruction on learners' vocabulary, grammar, and writing. |
| (*Amer, 2014*) | 2014 | Idioms | Not Available | Not Available | This research described the use of mobile applications for idiomatic expressions among language learners. |
| (*Petersen, Procter-Legg & Cacchione, 2013*) | 2013 | English language learning | Multiple Intelligence theory | Not Available | The ideas of mobility and creativity using LingoBee explored to support language learners. |
| (*Tsai et al., 2016*) | 2017 | English language learning | Experiential learning theory | Experiential learning theory used as a learning model | The experiential learning theory was used as a learning model because players seek knowledge by doing activities in the game environment. |
| (*Kim, Ruecker & Kim, 2017*) | 2017 | English language learning | Theory of diffusion | Not Available | The purpose of the study was to investigate the assistance of learning through mobile technology from students. |
| (*Wang, Zou & Xing, 2014*) | 2014 | Vocabulary | Autonomous learning theory | Not Available | This research trialed the use of mobile learning lexical spreadsheets for new vocabulary reference and consolidation. |
| (*Teng, 2018*) | 2019 | Vocabulary | Dual-coding theory | Working memory model and Multimedia principle | Dual-coding theory, working memory model, and Multimedia principle used in the study to investigate the effects of the captioning condition. |
| (*Botero, Questier & Zhu, 2018*) | 2019 | English language learning | Not Available | Self-regulated learning framework by Garrison | Self-directed learning, Garrison's comprehensive theoretical approach (1997) was used in this research. |
| (*Yurdagül & Öz, 2018*) | 2018 | English language learning | Not Available | Not Available | The study investigated the higher education students' attitude towards using the smartphone for language learning. |

(Continued)

| Ref. | Year | Application | Theory | Model/framework | Description |
|---|---|---|---|---|---|
| (Gafni, Achituv & Rahmani, 2017) | 2017 | English language learning | Unified theory | A unified theory of acceptance and use of technology | The study explored the usage of MALL applications impacts the attitude of learning for language learning. |
| (Zhang, 2016) | 2016 | Oral English learning | Not Available | Not Available | An application designed for practicing English orally for its users. |
| (Uematsu, 2012) | 2012 | English language learning | Not Available | Not Available | The researcher made the investigation of English as a foreign language effect in this study of Japan. |
| (Pham, Nguyen & Chen, 2017) | 2018 | English language learning | Not Available | Not Available | A mobile-based application, 'English Practice' was used to analyze active users' behavior from different countries. |
| (Çakmak & Erçetin, 2017) | 2018 | Vocabulary | A generative theory of multimedia learning | Not Available | The effect of multimedia glosses on vocabulary learning was investigated in this study. |
| (Shih, 2017) | 2017 | English for a specific purpose | Audiolingual theory | A framework for LINE application | For business language testing service (BULATS), a framework based on components were adopted: Teaching Method, Learning Satisfaction, Learning Effectiveness, and Qualitative analysis |
| (Ou-Yang & Wu, 2016) | 2017 | Vocabulary | Theory of cognitive style | Mobile app framework | An offline mobile learning system framework was developed, which worked any time and without the internet for vocabulary learning. |
| (Quan, 2016) | 2016 | Vocabulary | Not Available | Data drove learning model | A mobile application, 'mobile DDL,' was designed and developed for learning English. |
| (Tragant et al., 2015) | 2016 | Vocabulary | Not Available | Not Available | This study focused on a group of school children for English language instruction in the fall term. |
| (Chen & Lin, 2018) | 2018 | English language learning | Not Available | A Pedagogical framework | Task-based language teaching (TBLT) used as a pedagogical framework |
| (Bouchaib, Ahmadou & Abdelkader, 2018) | 2018 | English language learning | Attribution theory | Theoretical Framework of attribution theory | The study used Weiner's attribution theory (1992) in the Framework that provided failure or success reasons or explanations for people in education. |
| (Hazaea & Alzubi, 2018) | 2018 | English language learning | Not Available | Not Available | WhatsApp and Google were used to access reading material and to interact with peers and instructors. |
| (Kohnke, Zhang & Zou, 2019) | 2019 | Vocabulary | Cognitive theory of multimedia learning | Not Available | The study examined the effect of enhancing undergraduate students' knowledge retention of vocabulary. |
| (Chen, Liu & Huang, 2019) | 2019 | Vocabulary | Not Available | Not Available | An application 'PHONE Words' designed with game-related and non-game related functions to measure students' perception performance. |
| (Kirsch & Izuel, 2017) | 2019 | English language learning | Sociocultural theory | Not Available | The study investigated autonomy for language learning in primary schools using the iTEO application. |
| (Kalogirou, Beauchamp & Whyte, 2017) | 2019 | Vocabulary | Not Available | Not Available | The researchers experimented with vocabulary teaching to young children through drama. |
| (Makoe & Shandu, 2018) | 2018 | Vocabulary | Not Available | Sarrab et. al model | The aim was to design and develop a mobile application to enhance teaching and learning vocabulary. |
| (Shahbaz & Khan, 2017) | 2017 | Vocabulary | Not Available | Not Available | The study examined the mobile application efficiency on teaching phrases. |

| Ref. | Year | Application | Theory | Model/framework | Description |
|---|---|---|---|---|---|
| (Fisser, Voogt & Bom, 2012) | 2013 | Vocabulary | Instructional theory | Theoretical Framework | The 'Word Score,' a serious game, was developed based on Bom (2011) by discussing the study's literature. |
| (Ng et al., 2020) | 2020 | English language learning | Not Available | Module Intervention Model | The study investigated the performance of learners using mobile devices by examining a guided learning approach. |
| (Bourekkache & Kazar, 2020) | 2020 | English language learning | Not Available | M-Learning System architecture | An architectural model was proposed in the study with three components; the Domain model, the Pedagogical model, and the learner model. |
| (Klimova & Polakova, 2020) | 2020 | Vocabulary | Not Available | Not Available | The perception of students was discussed for the use of mobile applications aimed at learning vocabulary. |
| (Önal, Çevik & Şenol, 2019) | 2019 | Tenses | Not Available | The Framework of mobile learning tool acceptance | The study aimed to determine the effectiveness of the SOS mobile game to teach tenses in English. |
| (Zhang & Pérez-Paredes, 2019) | 2019 | English language learning | Second language acquisition theory | A theoretical model | By discussing two frameworks: 1) Framework for the rational analysis of mobile education (FRAME), and 2) TAM and TAM 2 model that focused on technology enhancement. |
| (Chu, Wang & Wang, 2019) | 2019 | Grammar | Not Available | Not Available | This study developed a mobile gaming approach using grammar concept mapping for English. |
| (Ramadoss & Wang, 2012) | 2012 | Grammar | Not Available | The affordance framework | The affordance framework, which consisted of pedagogical, social, and technological components, was used to examine the ICT tools affordance for language learning. |
| (Ishaq et al., 2020a) | 2020 | Usability | Not Available | Not Available | The usability was measure through a questionnaire survey for a mobile application. |
| (Ishaq et al., 2020b) | 2020 | Usefulness | Not Available | TAM | A survey was conducted based on the TAM model to see the usefulness of mobile applications. |
| (Ishaq et al., 2019) | 2019 | Effectiveness | Not Available | TAM | To measure the effectiveness of mobile application a survey was conducted from teachers and students. |
| (Ishaq et al., 2020d) | 2020 | Usability and Language Learning | Not Available | Not Available | An evaluation is conducted through questionnaires and interviews to measure the usability and design issues of a mobile application. |
| (Yanes & Bououd, 2019) | 2019 | English language learning | Not Available | Not Available | A taxonomy of challenges and opportunities produced by games for English language learning was provided. |
| (Angelia & Suharjito, 2019) | 2019 | English language learning | Not Available | Mechanics, Dynamics, and Aesthetics framework (MDA) | The aim of this study to enhance motivation and outcomes for English language learning using MDA framework. |
| (Tshering et al., 2018) | 2018 | Learning Spelling | Not Available | Not Available | An application 'Dzongkha App' developed for learning spelling for kids. |
| (Krasulia & Saks, 2020) | 2020 | English language learning | Behaviorist learning theory | Not Available | This study examined the perception and support provided using mobile technology from language learners. |

(Near-term/Long-term usefulness), Perceived Ease of use, Personal Innovativeness, and Behavioral Intention. This model was assessed based on data collection from 230 participants using a survey questionnaire (Liu, Li & Carlsson, 2010). A paradigm for the Mobile English Learning Objective (MELO), in which learners' understanding of mobile technologies may be directly correlated with three objects: Playfulness, Resistance to

change, and Self-management that were adapted after discussing extensive literature (*Huang et al., 2011*).

The M-COPE framework supported teachers to build mobile learning experiences using the ADDIE model. This Framework allowed teachers to understand five main aspects of mobile learning: mobile affordance, Ethics, Pedagogy, and Outcomes. It was validated by five experts (*Dennen & Hao, 2014*). The computational model concept and their proposed connections resulting from *Davis (1989)* TAM were introduced in which perceived ease of use, perceived usefulness, and perceived contextual variables were measured. This model was validated through SmartPLS 2.0 (*Bohm & Constantine, 2016*). A motivational model, ARCS (Attention, Relevance, Confidence, and Satisfaction) that considered learners' motivation and procedural assistance in compliance to demand, was used to measure effectiveness and learning motivation. This model has three features: (1) emphasize motivation and emotion stimulation, (2) integrated with other theories and design, and (3) enhances the instruction effects and learning process (*Wu, 2018*). A mobile English learning continuance intention framework was used. TAM (*Davis, 1989*) was adopted to see the influence of mobile learning English satisfaction with self-management of learning to measure Perceived Usefulness, Perceived Playfulness, and Resistance to change. The internal consistency and reliability were measured through the PLS-SEM application (*Huang, 2014*).

The study adopted the TAM by incorporating perceived convenience and curiosity factor for mobile learning. To examine the validity of the measurement model, Confirmatory Factor Analysis (CFA) was performed with SmartPLS (*Chang et al., 2013*). In contrast, experiential learning theory was used because players seek knowledge by doing activities in the game environment. Moreover, the learning process was exciting, challenging, and relevant, along with substantial experience provided to the players where learner motivation and effectiveness were measured (*Tsai et al., 2016*). In a dual-coding theory, the working memory model and multimedia principle used for pedagogy in this study were validated by doing an extensive literature review (*Teng, 2018*). In this research, self-directed learning, Garrison's comprehensive theoretical approach of (1997), was used to measure self-directed learning (motivation, self-management, and self-monitoring) (*Botero, Questier & Zhu, 2018*) whereas 'English Fun Dubbing' application was designed for practicing English orally, and evaluation of the application's effectiveness (convenience, flexibility, user-friendliness, rich material, language context) was carried out by *Zhang (2016)*. An application was developed to observe the multimedia glosses' effect on second language listening comprehension and vocabulary learning in a mobile learning environment. Its effectiveness was measured by the author (*Çakmak & Erçetin, 2017*).

Business language testing service (BULATS) adopted a framework based on Teaching Method, Learning Satisfaction, Learning Effectiveness, and qualitative analysis to measure satisfaction and attitude (*Shih, 2017*). 'MyEVA Mobile' is a mobile-based application developed to explore the learning attitude, learning achievements, learning styles, and university students' strategies to improve the students' vocabulary that worked any time and without the internet. The framework components are 'Smartphone client,' 'Wireless device,' and 'Log Analyzer Server.' Learning behavior, Perpetual learning styles, and

Knowledge proficiency were measured in this study (*Ou-Yang & Wu, 2016*). In another study (*Chen & Lin, 2018*), task-based language teaching (TBLT) was used as a pedagogical framework to measure technology-mediated TBLT. 'Weiner attribution theory (1992) was used as a framework that provided failure or success reasons or explanations for people in education (*Bouchaib, Ahmadou & Abdelkader, 2018*). Moreover, WhatsApp and Google were used to enhance the reading ability to measure learner autonomy (*Hazaea & Alzubi, 2018*) and retain business vocabulary, whereby an application 'Excel@EnglishPolyU' was developed and tested with undergraduate students (*Kohnke, Zhang & Zou, 2019*). To design and implement a mobile-based English vocabulary application for distance learning students in South Africa, 'VocUp' was developed, and its evaluation measures usability, scalability, reliability, and flexibility (*Makoe & Shandu, 2018*).

A theoretical framework for 'Word Score' serious game was developed based on *Bom (2011)* by exploring the literature to measure this study's motivation (*Fisser, Voogt & Bom, 2012*). In contrast, an architectural model was proposed in the study. The Pedagogical model, and learner model, were used by pilot testing (*Bourekkache & Kazar, 2020*), whereas a mobile game for tenses' SOS Table' was developed. The evaluation measured its effectiveness, user's motivation, acceptance, and attitude within the Framework of mobile learning tools (*Önal, Çevik & Şenol, 2019*). Moreover, by discussing two frameworks: (1) Framework for a rational analysis of mobile education (FRAME) and (2) TAM and TAM 2, the author proposed a new theoretical model that focuses on the technology enhancement, non-formal learning setting, and learner-centered to develop a better understanding for EFL learners on mobile English learning resources (MELR) in English language learning for Chinese postgraduate students (*Zhang & Pérez-Paredes, 2019*). Furthermore, it was also a social cognitive theory of motivation. Simultaneously, the affordance framework consisted of pedagogical, social, and technological components, used to examine the ICT tools affordance for language learning (*Ramadoss & Wang, 2012*). TAM was used in the study to measure perceived usefulness and ease of use (*Ishaq et al., 2020b*).

Summarize the Table 11, researchers developed applications and games for language learning in which majority of these developed for English Language Learning and vocabulary. Similarly, less application and games were developed for pedagogy (*Kearney et al., 2012*) (*Dennen & Hao, 2014*) reading (*Hsu, Hwang & Chang, 2013*) (*Lin, 2014*), pronunciation (*Liakin, Cardoso & Liakina, 2014*) idioms (*Wong, 2013*) (*Amer, 2014*) tenses (*Önal, Çevik & Şenol, 2019*) grammar (*Chu, Wang & Wang, 2019*; *Ramadoss & Wang, 2012*) spellings (*Tshering et al., 2018*) achievement, attitude, usefulness, usability and effectiveness (*Martin & Ertzberger, 2013*) (*Liu, Li & Carlsson, 2010*) (*Ishaq et al., 2020a*; *Ishaq et al., 2020b*; *Ishaq et al., 2019*; *Ishaq et al., 2020d*). From the selected studies, various models/frameworks were used as a base for developing an application in which TAM (*Hsu, Hwang & Chang, 2013*) (*Lin, 2014*) (*Chang et al., 2013*) (*Ishaq et al., 2020b*; *Ishaq et al., 2019*) mostly used. Many applications did not use any model or Framework as the base for the purpose.

### RQ3. What are different application domains for the MALL application, and in which various forms are these applications exposed for the end-users?

Several tools have been designed and developed by the researchers in MALL to support the students and teachers in learning the English language. A more in-depth analysis presenting that most of these tools and applications are student-centric and focus on learning (*Ishaq et al., 2020b*). While some of them support teachers in teaching and students in learning, thus can be categorized as a teaching and learning tool (*Ishaq et al., 2020b*). On the other hand, very few tools focus only on teaching (*Ramadoss & Wang, 2012*). Lastly, another emerging trend has been observed where these tools are augmented with gamification to make them more exciting and useful for the stakeholders (*Chu, Wang & Wang, 2019*). Table 12 presents these tools in different categories above.

#### *Learning aspect*

A mobile adaptive language learning system in which recommended material for reading and reading annotation facilities was provided to pupils to improve learning outcomes (*Hsu, Hwang & Chang, 2013*). A Mobile learning tool and five-step vocabulary learning (FSVL) strategy were developed in a situational learning environment to assess their effects on English as a foreign language (EFL) performance and learning motivation. According to the mobile learning tool, a global positioning system (GPS) was used to develop and provide learning material (*Huang et al., 2016*). Similarly, a mobile learning application 'Raz-Kids' for an extensive reading program was designed by 'Learning A to Z.' The tool aimed to create the concentration of the students to enhance their essential reading ability. Furthermore, it provides e-books of alphabets for students and teachers' class management tools (*Lin, 2014*).

The study conducted experimental research using ASR, Non-ASR and Control Group. The ASR group Nuance Dragon Dictation application was designed and installed on their mobile phone to finish weekly pronunciation activities with immediately written visuals (Pictures) feedback provided by the application without any human interaction. Five native French speakers pronounced all words and phrases to test the application (*Liakin, Cardoso & Liakina, 2014*). In a study by *Chang et al. (2013)*, 'Mebook,' a multimedia eBook system that can be played in the format of MP3 was developed. It integrated text, images, voice, and pictures that provided directions for listening, speaking, writing, and reading.

Furthermore, it also included speed adjustment of audio and switching of language from English to Chinese and Chinese to English. Duolingo, a free mobile assisted language learning tool (for vocabulary acquisition) is available in the web and mobile (Android and iOS) versions to examine out of class engagement informally. Translation exercises of sentences and words are the delivery method of this application. Moreover, instructors can see logs and progress to keep an eye on the students (*Gafni, Achituv & Rahmani, 2017*). A mobile-based gaming approach for English grammar and vocabulary learning 'Save the princess with Teddy' was developed and students' learning achievement with gameplay at different levels was analyzed. It was hoped to enhance students' learning motivation with the game and assistance to learn the English grammar concept (*Chu, Wang & Wang,*

**Table 12 Tools proposed by current studies.**

| | Ref. | Year | Application name | Description | Application evaluation method | Data set |
|---|---|---|---|---|---|---|
| **Learning aspect** | (*Hsu, Hwang & Chang, 2013*) | 2013 | Vocabulary and Annotation | A mobile adaptive language learning system provided to pupils for the improvement of learning outcomes. | An application developed based on the Literature Review | Not Available |
| | (*Zhang, Song & Burston, 2011*) | 2011 | 'Fetion' SMS text message | To explore the effectiveness of vocabulary learning through mobile technology, 'Fetion' an SMS text message application provided by the Chinese mobile. | Not Available | Not Available |
| | (*Huang et al., 2016*) | 2016 | Mobile Learning Tool | A mobile learning tool and five-step vocabulary learning (FSVL) strategy were developed in a situational learning environment to assess their effects on English as a foreign language (EFL) performance and learning motivation. | Developed based on the Literature Review | Not Available |
| | (*Lin, 2014*) | 2014 | Raz-kids (Online reading tool) | Raz-Kids, a mobile learning application for an extensive reading program developed by 'Learning A to Z' was used to enhance their essential reading ability. | This application is developed by Learning A to Z (www.learninga-z.com) | Not Available |
| | (*Liakin, Cardoso & Liakina, 2014*) | 2015 | Automatic Speech Recognition (ASR) | Nuance Dragon Dictation application was designed to finish weekly pronunciation activities with immediate written visuals (Pictures) feedback provided by the application without any human interaction. | Five native French speakers pronounced all words and phrases to test the accuracy of the application. | Not Available |
| | (*Bohm & Constantine, 2016*) | 2016 | CoLaLe app | The location-based vocabulary learning activities application for Thai and German users was proposed in the study to motivate personalized learning. | For evaluation of the application, a Short video was made that demonstrated the key features mentioned above to evaluate it using an online questionnaire. | Not Available |
| | (*Chang et al., 2013*) | 2013 | Mebook | Mebook, designed and developed by a Taiwanese company, is a multimedia eBook system and can be played in MP3. | Not Available | Not Available |
| | (*Petersen, Procter-Legg & Cacchione, 2013*) | 2013 | LingoBee | A crowdsourced mobile-based application, 'LingoBee,' which was open source and available freely, was explored in this study for creativity and mobility in language learning. | Evaluated through user studies in European Countries. | Not Available |

(Continued)

| Ref. | Year | Application name | Description | Application evaluation method | Data set |
|---|---|---|---|---|---|
| (*Petersen, Procter-Legg & Cacchione, 2013*) | 2013 | LingoBee | A crowdsourced mobile-based application, 'LingoBee,' which was open source and available freely, was explored in this study for creativity and mobility in language learning. | Evaluated through user studies in European Countries. | Not Available |
| (*Botero, Questier & Zhu, 2018*) | 2019 | Duolingo | Duolingo, a free mobile assisted language learning tool (vocabulary acquisition) available for web and mobile (Android and iOS) versions to examine out of class engagement informally. | Free application for Android and iOS | Duolingo dashboard |
| (*Gafni, Achituv & Rahmani, 2017*) | 2017 | Duolingo | Duolingo, a free mobile assisted language learning tool (vocabulary acquisition) available for web and mobile (Android and iOS) versions to examine out of class engagement informally. | Free application for Android and iOS | Duolingo dashboard |
| (*Pham, Nguyen & Chen, 2017*) | 2018 | English Practice | The application 'English Practice' was used to analyze the usage behavior of 53,825 users from 12 countries. | Online evaluation through the "Google Firebase analytics tool." | Logfile of Google Firebase |
| (*Çakmak & Erçetin, 2017*) | 2018 | Mobile-assisted listening application | 'Mobile assisted listening application' was developed to examine L2 multimedia glosses' effects in a mobile learning environment for vocabulary and listening comprehension. | Pilot study | Free recall method for listening comprehension |
| (*Ou-Yang & Wu, 2016*) | 2017 | MyEVA Mobile | An offline mobile assisted language learning application system, 'My English Vocabulary Assistant mobile edition' (MYEVA Mobile) was developed to engage mixed-modality vocabulary learning to improve their vocabulary. | Not Available | Logfile obtained from log analyzer server to analyze the results |
| (*Quan, 2016*) | 2016 | AKWIC | In the context of academic English, a mobile data-driven learning (DDL) experiment was discussed by designing and developing an application' AKWIC (academic key words in context)' for voluntary participants. | Not Available | Automatic logging to gather data from the application |
| (*Hazaea & Alzubi, 2018*) | 2018 | WhatsApp and internet search engines (Google) | Students used mobile applications like WhatsApp and Internet search engine (Google) to access reading and interaction materials with peers and instructors outside the class. | Not Available | The data is taken from the application& manually is considered one unit. |

| | Ref. | Year | Application name | Description | Application evaluation method | Data set |
|---|---|---|---|---|---|---|
| | (Chu, Wang & Wang, 2019) | 2019 | Save the princess with Teddy | A mobile-based gaming approach for English grammar and vocabulary learning 'Save the princess with Teddy' was developed and analyzed students' learning achievement with gameplay at different levels. | Not Available | Not Available |
| **Teaching and learning aspect** | (Kearney et al., 2012) | 2012 | Mobagogy & The Bird in the Hand | 'Mobagogy' a professional learning community of academicians to investigate to use of mobile technologies in learning and teaching whereas, 'The bird in the Hand' initiative by UK to examine the experience of trainee and newly qualified teachers by providing them smartphones to use in their offices and teaching schools to enhance professional practices. | Mobagogy developed by Australian University, UK sponsors the bird in the Hand | Not Available |
| | (Zhang, 2016) | 2016 | English Fun Dubbing | To evaluate mobile applications' benefits, an application 'English Fun Dubbing' is designed for the oral practice of English for its users. | Developed by a Chinese Sci-Tech company | Not Available |
| | (Shih, 2017) | 2017 | LINE app | LINE app, an application that was used to investigate the teaching English for specific purpose (ESP) effects on Business Language Testing Service (BULATS) after the class. | Not Available | Not Available |
| | (Makoe & Shandu, 2018) | 2018 | VocUp | To enhance English vocabulary teaching and learning, a mobile-based application 'VocUp' was designed and implemented. | Evaluated by external parties | Not Available |
| | (Shahbaz & Khan, 2017) | 2017 | WhatsApp | A mobile application, 'WhatsApp' was used in this study to examine the efficiency of the teaching of 40 phrases. | Not Available | Not Available |
| | (Klimova & Polakova, 2020) | 2020 | Anglictina (English) TODAY | An application Anglictina (English) TODAY was developed to help pupils prepare for final exams by learning from anywhere at any time. | Not Available | Not Available |
| | (Ramadoss & Wang, 2012) | 2012 | Grammar Grabber | To evaluate grammar at the school level, an online assessment tool, 'Grammar Grabber' was discussed in this study. | Not Available | Log files of user accounts |

(Continued)

| | Ref. | Year | Application name | Description | Application evaluation method | Data set |
|---|---|---|---|---|---|---|
| | (Ishaq et al., 2020b) | 2020 | Literacy and Numeracy Drive | An application, 'Literacy and Numeracy Drive' to teach and learn languages at the school level, was discussed. This application provided some practice and then some assessment exercises for the English language. | Not Available | Not Available |
| Gamification based learning and teaching | (Sandberg, Maris & Geus, 2011) | 2011 | MEL Application (Serious Game) | A serious game, 'MEL application,' developed in which 25 animals from five continents (Asia, South-America, Africa, Oceania, and North-America) were involved and played during the visit to the zoo home. | The pre-test and Post-test consisted of a vocabulary test to measure active and passive word knowledge. | Not Available |
| | (Holden & Sykes, 2011) | 2011 | Mentira | Augmented reality game-based language learning of Spanish in Southwestern US explores foreign and second language learning's benefits and complexities. | The interview was taken after playing this game by the players | Not Available |
| | (Sandberg, Maris & Hoogendoorn, 2014) | 2014 | Mobile English Learning2 (MEL2) (Game features added) | To investigate a mobile English learning application that would be supplement and support for learning English at school was adopted for the study by enhancing game features. | Adapted from Literature | An online database to store data |
| | (Wu, 2018) | 2018 | English Vocabulary Practice System Game | The mobile-based English Vocabulary system for practice was designed and developed to provide students with assistance to review, proficiency, and practice in the class and after the class. | This game system was developed based on the ARCS model. | Not Available |
| | (Amer, 2014) | 2014 | Idiomobile (Game for idioms) | 'Idiomobile' a mobile-based game that was made available for specific handsets based on the knowledge of using idiomatic expressions. | Not Available | Data collected from phones |
| | (Tsai et al., 2016) | 2017 | Game-based Happy English Learning System | A game-based system, 'Happy English Learning System', in which learning material was integrated to experiment to see the learner's motivation for achievement. | Not Available | Not Available |
| | (Kohnke, Zhang & Zou, 2019) | 2019 | Excel@EnglishPolyU (Alphabet vs. Aliens and Books vs. Brains@PolyU) | A game-based applications Excel@EnglishPolyU, Alphabet vs. Aliens, and Books vs. Brains@PolyU, in which business vocabulary after completing level challenges could acquire by the learners. | Not Available | Not Available |

| Ref. | Year | Application name | Description | Application evaluation method | Data set |
|---|---|---|---|---|---|
| (*Chen, Liu & Huang, 2019*) | 2019 | PHONE Words | An application 'PHONE Words' was developed for vocabulary learning, designed with game functions and without game functions to investigate the perceptions and measure learners' learning performance. | Not Available | Log File from the DB server in which usage behavior is recorded. |
| (*Fisser, Voogt & Bom, 2012*) | 2013 | Word Score | A serious online game was designed to extend the students' vocabulary in primary school was discussed in this study. | This game was part of the national project 'Educational Time Extension (ETE)'. | Not Available |
| (*Önal, Çevik & Şenol, 2019*) | 2019 | SOS Table | 'SOS Table,' a mobile-based game application, was designed and developed to repeat tenses, high-frequency words, and sentence string. | Not Available | Not Available |

*2019*). In the context of academic English, a mobile data-driven learning (DDL) experiment was discussed by designing and developing an application' AKWIC (academic key words in context)' for voluntary. This application provided support for academic writing (vocabulary learning), and data were obtained from logs generated by the application (*Quan, 2016*).

An SMS text message application, 'Fetion' explores the effectiveness of vocabulary learning using mobile technology. Specifically, SMS from mobile phones enhanced English vocabulary than outdated printed material (*Zhang, Song & Burston, 2011*). In contrast, a location-based vocabulary learning activities application for Thai and German users was proposed in a study with personalized learning motivation. Moreover, the aim was to enhance user acceptance by providing relevant learning content to evaluate an online questionnaire (*Bohm & Constantine, 2016*). A free mobile assisted language learning tool (vocabulary acquisition) 'Duolingo,' available for web and mobile (Android and iOS) versions examined out of class engagement informally. Translation exercises of sentences and words are the content of this application, and instructors can see logs and progress to track students' progress (*Botero, Questier & Zhu, 2018*). 'Mobile assisted listening application' was developed to examine L2 multimedia glosses' effects in a mobile learning environment, for vocabulary and listening comprehension, validated through a pilot study (*Çakmak & Erçetin, 2017*).

An offline mobile assisted language learning application system, 'My English Vocabulary Assistant mobile edition' (MYEVA Mobile) was developed to engage students in mixed-modality vocabulary learning to improve their vocabulary. The users learned targeted words via four vocabulary learning strategies: flashcard, imagery, word card, and Chinese assonance. A log analyzer server was installed, and log files were used to analyze the results (*Ou-Yang & Wu, 2016*). Furthermore, a mobile application like WhatsApp and Internet search engine (Google) were used by students to access materials for reading

and interaction with peers and instructors outside the class. This application was used to improve learning skills in EFL reading (*Hazaea & Alzubi, 2018*).

A crowdsourced mobile-based 'LingoBee,' an open-source and freely available application, explored creativity and mobility in language learning. It was the part of the European countries' project "SIMOLA", designed to help collaborative language learning using the idea from social networks and crowdsourcing. Users created content for language learning, stored it on their device, and shared it with other users via a cloud-based repository. The application was evaluated through user studies in European countries (*Petersen, Procter-Legg & Cacchione, 2013*). In contrast, an application 'English Practice' was used to analyze the usage behavior of 53,825 users from 12 countries, and the evaluation was conducted through log files generated by the Google Firebase analytics tool (*Pham, Nguyen & Chen, 2017*).

### Teaching and learning

An online assessment tool, 'Grammar Grabber,' was discussed in this study to evaluate grammar at the school level. Using this tool, users do practice online and get constant feedback to know their performance. The authors investigated this online assessment tool to perceive pedagogical, technological, and social affordances using multiple-choice questions and allowing re-attempt for wrongly answered questions. The log files of the application were used as the dataset (*Ramadoss & Wang, 2012*). An application, 'Literacy and Numeracy Drive' to teach and learn languages, was discussed at the school level. This application provided some practice and then some assessment exercises for the English language. This application aimed to teach singular/plural, use of has/have/had/is/am/are/was/were, and reading comprehension (*Ishaq et al., 2020b*). A serious online game that was designed to extend the students' vocabulary in primary school was discussed in this study. This game was part of the national project 'Educational Time Extension (ETE).' The students spend more time than expected class timing to learn and improve their vocabulary. Applications for teaching and learning at the college level was not adopted/developed for English language learning in the selected studies.

LINE App, an application that was used to investigate the teaching of English for specific purpose (ESP) effects on Business Language Testing Service (BULATS) (*Shih, 2017*). Whereas 'Mobagogy' a professional learning community of academicians to investigate the use of mobile technologies in learning and teaching while 'The bird in the Hand' initiative by the UK to examine the experience of trainee and newly qualified teachers by providing them smartphones to use in their offices and teaching schools to enhance professional practices of pedagogy. Australian University developed the application Mobagogy', and The Bird in the Hand was sponsored by the UK (*Kearney et al., 2012*) to evaluate the benefits of mobile applications and an application 'English Fun Dubbing' designed for English's oral practice its users. Learning material in this application was: animations, songs, movies, short videos, and textbooks developed by the Chinese Sci-Tech Company (*Zhang, 2016*). To enhance teaching and learning of English vocabulary, a mobile-based application 'VocUp' was designed and implemented to

evaluate external parties (*Makoe & Shandu, 2018*). A mobile application, 'WhatsApp' was used in this study to examine the efficiency of teaching 40 phrases.

Furthermore, the study also aimed to enhance vocabulary acquisition (*Shahbaz & Khan, 2017*). An application 'Anglictina (English) TODAY' was developed by a Ph.D. student of Computer Science with a language teacher's help. This application helped pupils for the preparation of final exams by learning from anywhere at any time. To discuss students' perception of using a mobile application, this was developed based on their needs to make EFL vocabulary teaching and learning useful (*Klimova & Polakova, 2020*).

### Gamification based learning and teaching

A serious game 'MEL application' developed in which 25 animals from five continents (Asia, South-America, Africa, Oceania, and North-America) were involved developed. The game was played during the visit to the zoo and at home in which two modes were available in which one was totally related to the zoo situation. One was independent of GPS that allowed children to access animals in different continents (*Sandberg, Maris & Geus, 2011*). To investigate a mobile English learning application that would be supplement and support for learning English at school was adopted for the study by enhancing game features. The application's learning material on animals from the zoo resulted in the students' outperformance using the mobile application for a fortnight than users for a fixed period (*Sandberg, Maris & Hoogendoorn, 2014*). A game-based system, 'Happy English Learning System,' in which learning material was integrated to experiment, was installed on mobile devices to see the learner's motivation for achievement. The activities were divided into levels of difficulties: Easy, Medium, and Advanced, according to the ability of users (*Tsai et al., 2016*).

Game-based applications Excel@ EnglishPolyU, Alphabet vs. Aliens, and Books vs. Brains@PolyU, business vocabulary after completing level challenges may acquire by the learners (*Kohnke, Zhang & Zou, 2019*). Similarly, an application, 'PHONE Words,' was developed by Alice English Education Studio for vocabulary learning, designed with game functions (MEVLA-GF) and without game functions (MEVLA-NGF) to investigate the perceptions and measure the learning performance of learners. This application also provides an assessment with a gamified competition mechanism (*Chen, Liu & Huang, 2019*). To give the students' opportunity, 'SOS Table,' a mobile-based game application, was designed and developed to repeat tenses, high-frequency words, and sentence string. Time challenge was provided to create negative, positive, or correct sentences with words and pronunciation after completing the task (*Önal, Çevik & Şenol, 2019*). The serious game was not adopted/developed at the college level for English language learning in the selected studies.

In the study (*Holden & Sykes, 2011*), an augmented reality-based language learning game developed for Spanish in Southwestern US to explore foreign and second language learning's benefits and complexities. Similarly, a mobile-based English Vocabulary system for practice was designed and developed not to replace traditional classroom teaching methods. The aim was to provide students with assistance to review, proficiency, and practice in and after the class. The game system was designed based on the ARCS model,

which provided assistance in vocabulary learning and listening and worked as a testing tool (*Wu, 2018*). 'Idiomobile' a mobile-based game that was made available for specific handsets based on the knowledge of using idiomatic expressions in a critical situation (*Amer, 2014*).

## RQ 4. What is the specific content adopted for teaching and learning in MALL research?

Teaching and learning through MALL by applying pedagogical skills or tools after adopting appropriate content are summarized and discussed in this section. The current section is divided into four sub-levels, i.e., Learning, Teaching and Learning, and Serious Game based presented in Table 13.

The content usage for learning aspect in MALL about reading perspective discussed for the undergraduate students of public university enrolled in "Computing and Information Technology (CIT)" adopted as curriculum (*Cheon et al., 2012*). Whilst personalized recommended learning material with reading annotation services were categorized into three difficulty levels: elementary, intermediated, and intermediated high level (*Hsu, Hwang & Chang, 2013*). In the study (*Bohm & Constantine, 2016*), discussed that multimedia learning material in the written and audio-visual form provided using the app according to the user's relevant current position. Words and pictures were adequate for learners to learn vocabulary by solving word pair quizzes. The content adopted in (*Chang et al., 2013*) was broad and varied, including current issues related to education, economy, environment, social, politics, technology, etc. For English learning, the topic areas covered were speaking, listening, reading, and vocabulary. To teach most popular idioms, content relevant to idioms and collocations was collected for application from books and websites (*Amer, 2014*).

A cooperative creation of multimedia content by adding it to new vocabulary is then stored on the devices and uploaded on the cloud by the users that can view or brows LingoBee repository by sorting it chronologically or alphabetically (*Petersen, Procter-Legg & Cacchione, 2013*). The study (*Tsai et al., 2016*) discussed the curriculum for English subject consisted of the topics: phonetic symbols, alphabet, grammar, phrases, vocabulary, and sentence patterns used by pupils to review it and also can share with the class by using social media (i.e., Facebook) whereas the course content in application 'Duolingo' focused variety of subject-specific vocabulary acquisition by learning nearly 2,000 words. The application's delivery techniques were translating from source to users' target language, choosing right translated phrases, paring of words from languages, flashcard practicing, and translation of unknown words (*Botero, Questier & Zhu, 2018*).

Teaching is a method of discussing and interfering with people's desires, perspectives, and emotions to learn specific things (*P. byinfed.org, 2020*). The reading perspective in teaching is discussed by *Ou-Yang & Wu (2016)* in which the content used for vocabulary teaching, suitable for 'Test of English for International Communication (TOEIC) that is a standardized test used in Taiwan. Fifty words were taken for vocabulary was chosen from TOEIC, including different difficulty levels.

**Table 13  Findings of the reviewed studies for content.**

| Criterion for content | Year/grade | Ref. | Evaluation method | Findings |
|---|---|---|---|---|
| Readiness of M-learning | 2012 (college) | (*Cheon et al., 2012*) | Structural Equation Modeling (SEM) | The participants were undergraduate students at a large, public university in which students were enrolled in a subject, "Computing and Information Technology (CIT)" which was the core curriculum subject required by all the students. |
| Reading annotation | 2013 (School) | (*Hsu, Hwang & Chang, 2013*) | Quasi-Experiment with the Technology Acceptance Model and Cognitive Load questionnaire | Personalized recommended learning materials with reading annotation services were provided to the students by analyzing profiles and portfolios using a developed adaptive learning system. The reading material was categorized into three difficulty levels: Elementary, Intermediate, and Intermediate-High level. |
| Written and Audio-Visual material | 2016 (University) | (*Bohm & Constantine, 2016*) | Partial least square structural equation modeling | Multimedia learning material in written and audio-visual form was provided using the CoLaLe app to enhance the vocabulary learning experience. The content was delivered using the app according to the relevant current position of the user. Words and pictures were useful for learners to learn vocabulary by solving word pair quizzes. |
| Reading and Listening material | 2013 (School) | (*Chang et al., 2013*) | One-group post-test design | The content in Mebook was broad and varied, including current issues related to education, economy, environment, social, politics, technology, etc. For English learning, the topic area covered was: speaking, listening, reading, and vocabulary. |
| Idioms collected from Books and Websites | 2014 (University) | (*Amer, 2014*) | The questionnaire, Application usage, and Exit Interview | Teaching the most popular idioms, content relevant to idioms and collocations was collected for application from books and websites. The idioms appeared in most websites, or books were selected into the application to teach ESLs. |
| Multimedia content added by the user | 2013 (Other) | (*Petersen, Procter-Legg & Cacchione, 2013*) | Conceptual Paper | A language learning application 'LingoBee' was designed for the cooperative creation of multimedia content by adding it to new vocabulary by the users. It was stored on the device of the users then uploaded to the cloud provided by the application. The content could be viewed or browsed LingoBee repository by sorting it chronologically or alphabetically. |
| Phonics, Symbols, Vocabulary, and Sentence pattern | 2017 (School) | (*Tsai et al., 2016*) | Quasi-experimental method | The English subject's curriculum consisted of phonetic symbols, alphabet, grammar, phrases, vocabulary, and sentence patterns. Pupils could use the HELS application to review the content and share the content prepared by them in the class and social media (i.e., Facebook). |
| Vocabulary acquisition | 2019 (University) | (*Botero, Questier & Zhu, 2018*) | Dashboard, Questionnaire, and Interview | The course content in application 'Duolingo' focused on various subjects, specifically vocabulary acquisition, by learning nearly 2000 words. The application's delivery techniques were translation from source to target language of users, choosing the right translated phrases, paring of words from languages, flashcard practicing, and translation of unknown words. |

| Criterion for content | Year/grade | Ref. | Evaluation method | Findings |
|---|---|---|---|---|
| Vocabulary teaching | 2017 (University) | (Ou-Yang & Wu, 2016) | Questionnaire, Pretest, Posttest, Log upload | The content used in this study for vocabulary teaching suitable for 'Test of English for International Communication (TOEIC), which is a standardized test used in Taiwan. Fifty words were taken for vocabulary was chosen from TOEIC, including different difficulty levels. |
| Animation and Videos for learning | 2016 (University) | (Zhang, 2016) | Anonymous online Questionnaire | An application designed by a Chinese Sci-Tech company to assist in English practice orally. It provided rich learning material of animations, movies, short videos, songs, and textbooks with English speakers speaking for different age group students. |
| Vocabulary learning | 2018 (University) | (Makoe & Shandu, 2018) | Interviews | For the vocabulary test, 'Word Capsules (short vocabulary tests)' was developed. Words (content) were selected from 10,000 words for the test, who gripped numerous words in English with a vast vocabulary and can survive with studying challenges at a higher level in English. |
| Vocabulary and Phrases learning | 2020 (University) | (Klimova & Polakova, 2020) | Questionnaire | Practicing and retaining new phrases and words while learning English, ten lessons of vocabulary and ten lessons of phrases were adopted, and pupils have to translate to English from the native language. |
| Pictures and Videos | 2011 (School) | (Sandberg, Maris & Geus, 2011) | Pre-test and Post-test | Pictures and videos to learn about zoo animals from five continents. |
| Classroom curriculum | 2011 (University) | (Holden & Sykes, 2011) | Interviews | To learn Spanish from an augmented reality (AR) game-based language learning tool, gameplay, narrative, setting, and curriculum were added. The classroom curriculum was adopted for the game for the three-to-four-week timeframe. |
| Vocabulary practicing | 2018 (University) | (Wu, 2018) | Questionnaire | A game-based English vocabulary practice system in which the vocabulary presentation model combined learning content with difficult vocabulary levels. |
| Vocabulary learning | 2019 (University) | (Chen, Liu & Huang, 2019) | Questionnaire and Interviews | For vocabulary learning using MEVLA-GF, interviewees' perception identified gamified design, interface design, and content design. The interface and gamified design efficiently be used, and content should also satisfy the learners' needs for different purposes and at the level for vocabulary learning. |
| Vocabulary, Sentences, Grammar | 2019 (School) | (Chu, Wang & Wang, 2019) | Quasi-experimental | Learning material consisted of vocabulary, sentence examples, grammar examples, and grammar concept mapping for English games in elementary school. The purpose was to make a strong foundation of English grammar and enhance its knowledge with the understandability of relationships between concepts. |

Current studies investigated the content usage according to speaking, listening, reading, and writing in teaching and learning through MALL. An application designed by the Chinese Sci-Tech company to assist in English practice orally provided rich learning material of animation movies, short videos, songs, and textbooks with English speakers

speaking for different age group students (*Zhang, 2016*). From the reading perspective, 'Word Capsules (short vocabulary test)' was developed in which words (content) were selected from 10,000 words, who gripped numerous phrases in English have a vast vocabulary and able to survive with challenges of studying at a higher level in English (*Makoe & Shandu, 2018*). In contrast, for all the viewpoints, practicing and retaining new phrases and words, ten lessons of vocabulary and ten lessons of phrases were adopted as content that the pupils have to translate to English from native language (*Klimova & Polakova, 2020*).

The content used in game-based applications for reading perspective investigated from selected studies in which the content was pictures and videos to learn about the zoo animals (*Sandberg, Maris & Geus, 2011*). Learning Spanish Augmented reality game-based language learning tool was also investigated in which classroom curriculum was adopted for the game for three to four-week timeframe (*Holden & Sykes, 2011*). For vocabulary practice through a game-based system, a presentation model combined learning content with difficulty levels (easy, medium, and advance) (*Wu, 2018*). Still, for vocabulary learning using MEVLA-GF, the perception of interviewees identified gamified design, interface design, and content design that satisfy the learners' need for different purposes and at level (*Chen, Liu & Huang, 2019*). In the study of *Chu, Wang & Wang (2019)*, learning material consisted of vocabulary, sentence examples, grammar examples, and grammar concept mapping for English games in elementary school. The purpose was to make a strong foundation of English grammar and enhance its knowledge with the understandability of relationships between concepts.

### RQ 5. How and in what different perspectives the MALL applications were evaluated; and what were the evaluation measures and tools used for their evaluation?

RQ5 was posed to investigate the evaluation methodologies and tools used by the MALL studies are presented in Tables 14 and 15. Table 14 shows that 33 out of 67 studies adopted or developed a questionnaire to collect data from the selected population. In contrast, only six studies used interviews and observations for their studies. The mixed-method approach in which questionnaires, interviews, and observations were used as a tool for collecting data by 18 studies while only five studies used discussion and other methods for their data collection.

### Evaluation

The evaluation in this section was teaching and learning and technical perspective. Researchers used various methods to collect data, and statistical tests were applied to analyze the data to produce relevant results. The statistical tests for analysis applied through various tools were, i.e., SPSS, PLS, etc. The methodology and tools used by the selected studies were presented in the following section.

**Table 14  Research methodologies adopted by studies.**

| Methodology | Instrument | Ref. sources | Total |
|---|---|---|---|
| Quantitative | Questionnaire | (*Cheon et al., 2012*) (*Martin & Ertzberger, 2013*) (*Liu, Li & Carlsson, 2010*) (*Sandberg, Maris & Geus, 2011*) (*Hsu, Hwang & Chang, 2013*) (*Wong & Looi, 2010*) (*Zhang, Song & Burston, 2011*) (*Sandberg, Maris & Hoogendoorn, 2014*) (*Huang et al., 2016*) (*Lin, 2014*) (*Huang et al., 2011*) (*Liu & Chen, 2014*) (*Liakin, Cardoso & Liakina, 2014*) (*Wong, 2013*) (*Wu, 2018*) (*Huang, 2014*) (*Chang et al., 2013*) (*Elgün-Gündüz, Akcan & Bayyurt, 2012*) (*Tsai et al., 2016*) (*Teng, 2018*) (*Zhang, 2016*) (*Çakmak & Erçetin, 2017*) (*Shih, 2017*) (*Kohnke, Zhang & Zou, 2019*) (*Kalogirou, Beauchamp & Whyte, 2017*) (*Shahbaz & Khan, 2017*) (*Fisser, Voogt & Bom, 2012*) (*Ng et al., 2020*) (*Klimova & Polakova, 2020*) (*Önal, Çevik & Şenol, 2019*) (*Chu, Wang & Wang, 2019*) (*Ishaq et al., 2020a*) (*Ishaq et al., 2019*) | 33 |
| Qualitative | Interview observation | (*Holden & Sykes, 2011*) (*Wang, Zou & Xing, 2014*) (*Uematsu, 2012*) (*Hazaea & Alzubi, 2018*) (*Kirsch & Izuel, 2017*) (*Makoe & Shandu, 2018*) | 6 |
| Mix method | Questionnaire interview observation | (*Dashtestani, 2015*) (*Lai & Zheng, 2017*) (*Avci & Adiguzel, 2017*) (*Amer, 2014*) (*Kim, Ruecker & Kim, 2017*) (*Botero, Questier & Zhu, 2018*) (*Yurdagül & Öz, 2018*) (*Gafni, Achituv & Rahmani, 2017*) (*Ou-Yang & Wu, 2016*) (*Quan, 2016*) (*Tragant et al., 2015*) (*Chen & Lin, 2018*) (*Bouchaib, Ahmadou & Abdelkader, 2018*) (*Chen, Liu & Huang, 2019*) (*Zhang & Pérez-Paredes, 2019*) (*Ramadoss & Wang, 2012*) (*Ishaq et al., 2020b*) (*Ishaq et al., 2020d*) | 18 |
| Other | Discussion others | (*Dennen & Hao, 2014*) (*Bohm & Constantine, 2016*) (*Petersen, Procter-Legg & Cacchione, 2013*) (*Pham, Nguyen & Chen, 2017*) (*Bourekkache & Kazar, 2020*) | 5 |

### Measures for teaching and learning

In the teaching and learning section, terminologies were described from the selected studies also presented the instruments used in these studies were shown in Table 15.

*Pedagogy:* a study of educational techniques, including the purposes of instruction and the approaches to accomplish them (*Peel, 2017*).

*Motivation:* originates from the term 'motive' that means needs, wants, wishes, or drives in people. It is the method of inspiring individuals to take steps to meet the targets (*MSG Management Study Guide, 2020*).

*Perception:* To organize, define, and interpret sensory input to reflect and recognize the input or situation presented (*Perception, 2020*).

*Curiosity:* originated from the flow concept, which means people prefer to communicate in the state of flow with their surroundings. Curiosity is preserved when people consider the world as fun or fascinating (*Chang et al., 2013*).

*Continuance Intention:* The level to which people intend to keep using smartphone English learning (*Chang et al., 2013*).

*Attitude:* An optimistic, pessimistic, or mixed assessment of an entity communicated at a certain level of anxiety. This represents an advantageous or disadvantageous assessment of a person, position, object, or event (*iEduNote.com, 2020*).

*Achievement:* Somebody succeeded in achieving something, particularly after much effort (*Collins English Dictionary, 2020*).

*Self-Directed Learning:* A learning approach that encourages learners to take control of their learning method (diagnosis requirements for learning, assessment of learning

**Table 15 Approaches used in applications/games of current studies.**

| Item no. | Game/application | Evaluation method | Approach measured | Tool to measure | Ref. |
|---|---|---|---|---|---|
| 1 | Mobagogy & The Bird in the Hand | Developed by Australian University and UK | Pedagogy | – | (Kearney et al., 2012) |
| 2 | MEL Application (Serious Game) | The pre-test and Post-test consisted of a vocabulary test to measure active and passive word knowledge. | – | Questionnaire | (Sandberg, Maris & Geus, 2011) |
| 3 | Vocabulary and Annotation | Literature Review | – | Questionnaire | (Hsu, Hwang & Chang, 2013) |
| 4 | Mentira | Interview was taken after playing this game by the players | – | Interview and Observation | (Holden & Sykes, 2011) |
| 5 | 'Fetion' SMS text message | – | – | – | (Zhang, Song & Burston, 2011) |
| 6 | Mobile English Learning2 (MEL2) (Game features added) | Adopted from Literature | – | – | (Sandberg, Maris & Hoogendoorn, 2014) |
| 7 | Mobile Learning Tool | Literature Review | Motivation | Questionnaire | (Huang et al., 2016) |
| 8 | Raz-kids (Online reading tool) | A to Z (Company) | Perception | Questionnaire | (Lin, 2014) |
| 9 | Automatic Speech Recognition (ASR) | Five native French speakers | – | – | (Liakin, Cardoso & Liakina, 2014) |
| 10 | CoLaLe app | A Short Video | Perceived usefulness, Perceived Ease of use | – | (Bohm & Constantine, 2016) |
| 11 | English Vocabulary Practice System Game | This game system developed on the basis of ARCS model. | Effectiveness and Motivation | Questionnaire | (Wu, 2018) |
| 12 | Mebook | – | Perceived usefulness, Perceived convenience, curiosity, and continuance intention | Questionnaire | (Chang et al., 2013) |
| 13 | Idiomobile (Game for idioms) | – | Attitude, Motivation | Questionnaire, Interviews | (Amer, 2014) |
| 14 | LingoBee | User Studies | Motivation and Achievement | Questionnaire | (Petersen, Procter-Legg & Cacchione, 2013) |
| 15 | Game-based Happy English Learning System | – | Motivation, Effectiveness | Questionnaire | (Tsai et al., 2016) |
| 16 | Duolingo | Free application for Android and iOS | Self-directed learning | Questionnaire, Interviews | (Botero, Questier & Zhu, 2018) |
| 17 | Duolingo | Free application for Android and iOS | Attitude | Questionnaire | (Gafni, Achituv & Rahmani, 2017) |
| 18 | English Fun Dubbing | Developed by Chinese Sci-Tech company | – | Questionnaire | (Zhang, 2016) |
| 19 | English Practice | Online evaluation | Behavior pattern | – | (Pham, Nguyen & Chen, 2017) |

(Continued)

| Item no. | Game/application | Evaluation method | Approach measured | Tool to measure | Ref. |
|---|---|---|---|---|---|
| 20 | Mobile-assisted listening application | Pilot study | Effectiveness | Questionnaire | (*Çakmak & Erçetin, 2017*) |
| 21 | LINE app | – | Satisfaction, Attitude | Questionnaire, Classroom observation | (*Shih, 2017*) |
| 22 | MyEVA Mobile | Literature Review | Attitude, Achievements, Learning behavior | Questionnaire | (*Ou-Yang & Wu, 2016*) |
| 23 | AKWIC | – | Attitude, Effectiveness | Questionnaire, Interviews | (*Quan, 2016*) |
| 24 | WhatsApp and internet search engines (Google) | – | Learner's autonomy | Interviews | (*Hazaea & Alzubi, 2018*) |
| 25 | Excel@EnglishPolyU (Alphabet vs. Aliens and Books vs. Brains@PolyU) | – | – | Online questionnaire | (*Kohnke, Zhang & Zou, 2019*) |
| 26 | PHONE Words | – | Perception | Questionnaire, Interviews | (*Chen, Liu & Huang, 2019*) |
| 27 | VocUp | Evaluated by external parties | Usability, Scalability, Reliability, and Flexibility | Interview | (*Makoe & Shandu, 2018*) |
| 28 | WhatsApp | – | Efficiency | – | (*Shahbaz & Khan, 2017*) |
| 29 | Word Score | This game was part of the national project 'Educational Time Extension (ETE)'. | Motivation, Effectiveness | Questionnaire, Interviews | (*Fisser, Voogt & Bom, 2012*) |
| 30 | Anglictina (English) TODAY | – | Perception | Questionnaire | (*Klimova & Polakova, 2020*) |
| 31 | SOS Table | – | Effectiveness | Questionnaire and Interviews | (*Önal, Çevik & Şenol, 2019*) |
| 32 | Save the princess with Teddy | Literature Review | – | Questionnaire | (*Chu, Wang & Wang, 2019*) |
| 33 | Grammar Grabber | – | Affordance | Questionnaire, Interviews | (*Ramadoss & Wang, 2012*) |
| 34 | Literacy and Numeracy Drive | – | Perceived usefulness, Ease of Use | Questionnaire, Interviews | (*Ishaq et al., 2020b*) |

priorities, choice of learning methods, and the measurement of academic achievement and outcomes) (*IGI Global, 2020*).

*Behavior Pattern:* A repeated way for a person or group to behave against a particular object or condition (*Dictionary.com, 2020*).

*Learning Behavior:* Learning Behavior stresses the crucial relationship between child and youth learning and their social experience and behavior (*Northampton Centre for Learning Behaviour, 2020*).

*Learner Autonomy:* The autonomy of learning is when pupils take care of their learning, both in respect of what they learn and, in the direction, they study it (*Oxford University Press ELT, 2013*).

*Affordance:* An affordance is an object quality or a condition that makes it possible for the person to take action (*Definitions for Affordance, 2020*).

### Technical evaluation measures

It is the Human-Computer Interaction (HCI) category to measure the applications' UI/UX (*Ishaq et al., 2019*). The following were the terminologies discussed in selected studies also, approaches used in these studies were shown in Tables 14 and 15.

*Usefulness:* The perceived usefulness (PU) is also one of the separate structures in the TAM. It is the extent to which a person believes that a specific process may improve the efficiency of his/her job (*World Leaders in Research-Based User Experience, 2020*) (*Ishaq et al., 2020c*).

*Ease of Use:* The primary use of computer programs in the TAM is a significant determinant of the target. A discreet individual describes a simple usage as evident in implementing a procedure and directly impacts the perceived usefulness (*Ishaq et al., 2020b*).

*Effectiveness:* Effectiveness is a participant's ability to execute a task in a given context. In general, efficacy is assessed by determining whether participants can carry out such tasks (*Harrison, Flood & Duce, 2013*).

*Perceived convenience:* The extent to which users perceive mobile English learning to be comfortable in terms of time, location, and the method to complete a task (*Chang et al., 2013*).

*Usability:* Usability is a common factor that determines how convenient it is to use interface design. The term "usability" also applies to the approaches used during the design process to increase ease of use (*World Leaders in Research-Based User Experience, 2020*).

*Efficiency:* Efficiency is the user's ability to deliver their role quickly and productively, representing the user's value during its use. Quality can be calculated in various ways, for example, the time taken to complete or the number of keystrokes necessary to finish a task (*Collins English Dictionary, 2020*).

*Scalability:* Scalability is characteristic of an entity, structure, model, or process, defining its ability, under increased or expanding workload or scope to handle and compete well (*Hayes, 2020*).

*Reliability:* Reliability means the possibility that a product, device, or service can perform its intended function properly for a given period or work without interruption within a fixed environment (*ASQ, 2020*).

*Flexibility:* Flexibility is a characteristic that explains how a person can tolerate changes in situations and think in novel, imaginative ways about issues and tasks (*Flexibility, 2020*).

### Evaluation methodologies

The methodology is the basic techniques or methods used to identify, collect, retrieve, and interpret information on the topic (*Paul, 2000*). A quantitative study using a quasi-experiment (Pre-test and Post-test) was conducted by (*Hsu, Hwang & Chang, 2013*) (*Wu, 2018*) (*Chang et al., 2013*) (*Kalogirou, Beauchamp & Whyte, 2017*) (*Fisser, Voogt & Bom,*

*2012*) to collect data through questionnaires whereas, a quantitative study using questionnaire only was conducted by (*Zhang, 2016*) (*Kohnke, Zhang & Zou, 2019*) (*Bourekkache & Kazar, 2020*). A mixed-method approach was used (*Amer, 2014*) (*Botero, Questier & Zhu, 2018*) (*Ou-Yang & Wu, 2016*; *Quan, 2016*) (*Önal, Çevik & Şenol, 2019*; *Zhang & Pérez-Paredes, 2019*) (*Ishaq et al., 2020b*) to conduct interviews and questionnaire for the collection of information, whereas a qualitative approach in which interviews were conducted by (*Hazaea & Alzubi, 2018*) (*Makoe & Shandu, 2018*).

*Statistical analysis*

Statistical analysis is data compilation and evaluation, allowing patterns and developments to be discovered (*Rouse, 2020*). The following were the statistical analysis techniques used by the selected studies:

*Mean and Standard Deviation:* Mean which is the average of data set (adding all the numbers then divided by its total point) (*Wei, 2020*) was calculated by *Liu, Li & Carlsson (2010)*, *Lin (2014)*, *Amer (2014)*, *Petersen, Procter-Legg & Cacchione (2013)*, *Çakmak & Erçetin (2017)*, *Ou-Yang & Wu (2016)*, *Quan (2016)* and *Fisser, Voogt & Bom (2012)* whereas Standard Deviation (SD) measures a dataset's dispersion relative to its mean and is calculated as the square root of the variance (*Hargrave, 2020*). It was calculated by *Wu (2018)*, *Botero, Questier & Zhu (2018)*, *Fisser, Voogt & Bom (2012)* and *Ishaq et al. (2020b)*.

*Analysis of Variance:* Analysis of variance (ANOVA) is a statistical method that evaluates a nominal level variable with two or more categories in a scale level dependent variable (*Statistics Solutions, 2013a*) was calculated by *Liu, Li & Carlsson (2010)*, *Hsu, Hwang & Chang (2013)*, *Lin (2014)*, *Çakmak & Erçetin (2017)* and *Kalogirou, Beauchamp & Whyte (2017)*.

*T-test:* The independent t-test is a method that contrasts two sets of a variable usually distributed on a mean value of a constant (e.g., interval or ratio) (*Statistics Solutions, 2013b*) was calculated by *Wu (2018)*, *Petersen, Procter-Legg & Cacchione (2013)* and *Önal, Çevik & Şenol (2019)*.

*Analysis of covariance:* Analysis of Covariance (ANCOVA) is the inclusion of a continuous variable in addition to the variables of interest (i.e., the dependent and independent variable) as means for control (*Statistics Solutions, 2013c*) calculated in *Hsu, Hwang & Chang (2013)* and *Ou-Yang & Wu (2016)*.

*Multivariate Analysis of Variance:* Multivariate Analysis of Variance (MANOVA) is similar to ANOVA, except that instead of one metric dependent variable, having two or more dependent variables and is concerned with examining the differences between groups (*Statistics Solutions, 2013d*) calculated by *Lin (2014)* and *Çakmak & Erçetin (2017)*.

*Linear Regression:* Linear regression is an analysis that assesses whether one or more predictor variables explain the dependent (criterion) variable (*Statistics Solutions, 2013e*) calculated by *Petersen, Procter-Legg & Cacchione (2013)*.

*Frequencies:* A frequency distribution is a graphical or tabular representation that indicates the number of observations over a given interval (*Young, 2020*) calculated by *Zhang (2016)*, *Kohnke, Zhang & Zou (2019)*, *Bourekkache & Kazar (2020)* and *Zhang & Pérez-Paredes (2019)*.

It is showing by Tables 14 and 15 that questionnaire tool was used to evaluate motivation (*Sandberg, Maris & Geus, 2011*), perception (*Hsu, Hwang & Chang, 2013*), effectiveness (*Huang et al., 2016*), usefulness (*Lin, 2014*), convenience (*Wu, 2018*), curiosity (*Chang et al., 2013*), achievement (*Petersen, Procter-Legg & Cacchione, 2013*), attitude (*Tsai et al., 2016*) (*Gafni, Achituv & Rahmani, 2017*), behavior (*Ou-Yang & Wu, 2016*), and perception (*Klimova & Polakova, 2020*), whereas interviews were conducted to measure the learners' autonomy (*Hazaea & Alzubi, 2018*), usability, scalability, reliability, and flexibility (*Makoe & Shandu, 2018*). Few studies used questionnaire and interview (both) to see the attitude and motivation (*Amer, 2014*), effectiveness (*Quan, 2016*) (*Önal, Çevik & Şenol, 2019*), perception (*Chen, Liu & Huang, 2019*), affordance (*Ramadoss & Wang, 2012*), usefulness, and ease of use (*Ishaq et al., 2020b*). In contrast, only (*Shih, 2017*) study used a questionnaire with class observation to measure students' satisfaction and attitude.

*Tools used for analysis*

Table 16 presented that 39 of the papers used statistical package for social sciences (SPSS) software which was of quantitative in nature, used to evaluate their research. Partial Least Squares (PLS) was also used by two (02) studies (*Huang, 2014*) (*Chang et al., 2013*).

Nvivo is a tool that was qualitatively used by *Wang, Zou & Xing (2014)* and *Quan (2016)*. After conducting interviews of their respondents, while only two (02) studies (*Klimova & Polakova, 2020*) (*Ramadoss & Wang, 2012*) evaluated their results manually to present in the articles. Mplus is a statistical modeling program that allows researchers to analyze the data used by *Cheon et al. (2012)*, whereas 'Wenjuan Wang' an online tool to analyze data used in China by *Zhang (2016)*. 'Facets,' 'Google Firebase,' 'Notes taking,' online tool, and 'Google Sheet' used by *Uematsu (2012)*, *Pham, Nguyen & Chen (2017)*, *Hazaea & Alzubi (2018)*, *Kohnke, Zhang & Zou (2019)* and *Ng et al. (2020)*, respectively. In this literature review, eleven (11) studies did not use any online or desktop application to analyze the data.

## RQ6: compare the usage of simple mobile applications with gamified applications (Serious Game) for language learning?

Figure 7 shows the trend of language learning applications and games developed during selected studies of 2010 to 2020. It can be observed that in the years 2010 and 2011, few applications related to mobile-based and gamified language learning applications have been proposed as fewer people were familiar with this area. From the year 2013 to 2014, an increasing number of gamified application gamified applications were proposed for the studies. Similarly, there are increments from 2017 to onwards for mobile-based gamified applications to learn how educational institutes were observed in the finalized studies.

Mobile and gamified language learning applications from the selected studies concerning level, purpose, evaluation method, and results are presented in Table 17. The applications are divided into two categories: (1) Mobile application, (2) Gamified applications (serious games). Mobile applications are proposed for teaching

**Table 16 Tools used for analysis in current studies.**

| Study Ref. | Tool | Total |
|---|---|---|
| (*Martin & Ertzberger, 2013*) (*Liu, Li & Carlsson, 2010*) (*Sandberg, Maris & Geus, 2011*) (*Hsu, Hwang & Chang, 2013*) (*Wong & Looi, 2010*) (*Zhang, Song & Burston, 2011*) (*Sandberg, Maris & Hoogendoorn, 2014*) (*Huang et al., 2016*) (*Lin, 2014*) (*Huang et al., 2011*) (*Dashtestani, 2015*) (*Liu & Chen, 2014*) (*Liakin, Cardoso & Liakina, 2014*) (*Wong, 2013*) (*Wu, 2018*) (*Lai & Zheng, 2017*) (*Avci & Adiguzel, 2017*) (*Elgün-Gündüz, Akcan & Bayyurt, 2012*) (*Amer, 2014*) (*Tsai et al., 2016*) (*Kim, Ruecker & Kim, 2017*) (*Botero, Questier & Zhu, 2018*) (*Yurdagül & Öz, 2018*) (*Gafni, Achituv & Rahmani, 2017*) (*Çakmak & Erçetin, 2017*) (*Shih, 2017*) (*Ou-Yang & Wu, 2016*) (*Tragant et al., 2015*) (*Chen & Lin, 2018*) (*Bouchaib, Ahmadou & Abdelkader, 2018*) (*Chen, Liu & Huang, 2019*) (*Shahbaz & Khan, 2017*) (*Fisser, Voogt & Bom, 2012*) (*Önal, Çevik & Şenol, 2019*) (*Chu, Wang & Wang, 2019*) (*Ishaq et al., 2020a*) (*Ishaq et al., 2020b*) (*Ishaq et al., 2019*) (*Ishaq et al., 2020d*) | Statistical Package for Social Sciences (SPSS) | 39 |
| (*Huang, 2014*) (*Chang et al., 2013*) | Partial Least Squares (PLS) | 2 |
| (*Wang, Zou & Xing, 2014*) (*Quan, 2016*) | Nvivo | 2 |
| (*Klimova & Polakova, 2020*) (*Ramadoss & Wang, 2012*) | Manually | 2 |
| (*Cheon et al., 2012*) | Mplus 6.11 | 1 |
| (*Zhang, 2016*) | Wenjuan Wang (www.wenjuan.com) | 1 |
| (*Uematsu, 2012*) | Facets 3.62 | 1 |
| (*Pham, Nguyen & Chen, 2017*) | Google Firebase analytics tool | 1 |
| (*Hazaea & Alzubi, 2018*) | Notes taking | 1 |
| (*Kohnke, Zhang & Zou, 2019*) | Online tool | 1 |
| (*Ng et al., 2020*) | Google Sheet | 1 |
| (*Kearney et al., 2012*) (*Holden & Sykes, 2011*) (*Dennen & Hao, 2014*) (*Bohm & Constantine, 2016*) (*Petersen, Procter-Legg & Cacchione, 2013*) (*Teng, 2018*) (*Kirsch & Izuel, 2017*) (*Kalogirou, Beauchamp & Whyte, 2017*) (*Makoe & Shandu, 2018*) (*Bourekkache & Kazar, 2020*) (*Zhang & Pérez-Paredes, 2019*) | Did not use analysis tool | 11 |

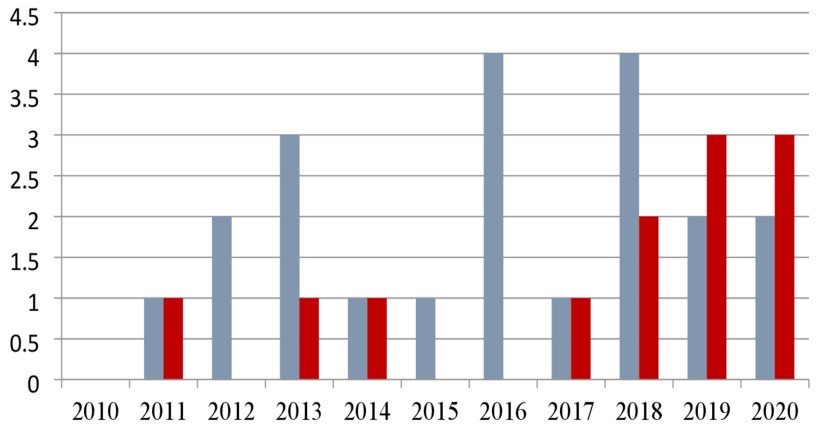

**Figure 7 Graph of MALL and Gamified Application.**

(*Kearney et al., 2012*) (*Zhang, 2016*), reading (*Hsu, Hwang & Chang, 2013*) (*Lin, 2014*) (*Hazaea & Alzubi, 2018*), vocabulary learning (*Zhang, Song & Burston, 2011*) (*Huang et al., 2016*) (*Bohm & Constantine, 2016*) (*Petersen, Procter-Legg & Cacchione, 2013*) (*Botero, Questier & Zhu, 2018*) (*Çakmak & Erçetin, 2017*) (*Ou-Yang & Wu, 2016*;

**Table 17 Mobile application and gamified applications.**

| Ref. | Year | Applications | Purpose | Level | Evaluation | Result |
|---|---|---|---|---|---|---|
| **Mobile Applications** | | | | | | |
| (Kearney et al., 2012) | 2012 | Mobagogy & The Bird in the Hand | Pedagogy | University | Not Available | Not Available |
| (Hsu, Hwang & Chang, 2013) | 2013 | Vocabulary and Annotation | Reading | School | Mean, SD, ANOVA, ANVOCA | Both experimental groups outperformed the control group but no difference in learning outcome between experimental groups. |
| (Zhang, Song & Burston, 2011) | 2011 | 'Fetion' SMS text message | Vocabulary learning | University | Mean and SD | Vocabulary learning through the proposed method is useful that may help to increase the effectiveness. |
| (Huang et al., 2016) | 2016 | Mobile Learning Tool | Vocabulary learning | School | Mean, SD and Paired Sample t-test, ANCOVA | Learning motivation and performance with mobile learning tools are superior to traditional learning. |
| (Lin, 2014) | 2014 | Raz-kids (Online reading tool) | Reading ability | School | Mean, SD, ANOVA, MANOVA | The mobile group outperformed the PC group in online activities and reading achievement. |
| (Liakin, Cardoso & Liakina, 2014) | 2015 | Automatic Speech Recognition (ASR) | Pronunciation | School | One-way ANOVA & t-test | The result showed that the learning environment is encouraging learning French through ASR. |
| (Bohm & Constantine, 2016) | 2016 | CoLaLe app | Vocabulary learning | University | Not Available | Students perceived that the proposed app is beneficial for learning. |
| (Chang et al., 2013) | 2013 | Mebook | Read, Write, Speak, and listen. | School | Not Available | The result showed that curiosity and perceived convenience has a positive effect on usefulness and continuance intention. |
| (Petersen, Procter-Legg & Cacchione, 2013) | 2013 | LingoBee | Vocabulary learning | Others | Mean, t-test, Linear regression | The result showed a significant difference and a positive impact on the experimental group. |
| (Botero, Questier & Zhu, 2018) | 2019 | Duolingo | Vocabulary acquisition | University | Mean, SD | The study concluded that MALL could engage university students in informal and out of class learning. |
| (Zhang, 2016) | 2016 | English Fun Dubbing | Pedagogy | University | Frequencies | Results showed satisfaction inconveniences, flexibility, user-friendliness, rich material, and authentic language context. |
| (Pham, Nguyen & Chen, 2017) | 2018 | English Practice | English practice | Others | Frequencies | Delivering learning content and designing the application interface shows better user behavior. |
| (Çakmak & Erçetin, 2017) | 2018 | Mobile-assisted listening application | Vocabulary and Listening comprehension | University | Mean, SD, ANOVA, MANOVA | The finding shows that access to glosses helped recognition. |
| (Ou-Yang & Wu, 2016) | 2017 | MyEVA Mobile | Vocabulary learning | University | Mean, SD, ANCOVA | Results revealed that mixed-modality vocabulary learning using MALL help EFL learners. |
| (Quan, 2016) | 2016 | AKWIC | Vocabulary learning | College | Mean | Results showed that DDL could be adapted in mobile devices to assist vocabulary learners. |
| (Hazaea & Alzubi, 2018) | 2018 | WhatsApp and internet search engines (Google) | Reading | University | Not Available | It can be concluded that learners have developed a sense of learner autonomy relevant to the choice of learning materials. |

(Continued)

| Ref. | Year | Applications | Purpose | Level | Evaluation | Result |
|------|------|-------------|---------|-------|-----------|--------|
| (*Makoe & Shandu, 2018*) | 2018 | VocUp | Vocabulary learning | University | Not Available | The importance of technology, pedagogy, mobile-app interventions highlighted in this paper. |
| (*Klimova & Polakova, 2020*) | 2020 | Anglictina (English) TODAY | Vocabulary teaching | University | Frequencies | Results indicate that the mobile app is facilitative for learning but not supportive of communication performance, with no pronunciation support and teachers' notifications. |
| (*Chu, Wang & Wang, 2019*) | 2019 | Save the princess with Teddy | Grammar and Vocabulary | School | Mean, SD, ANCOVA | The result shows the higher significant achievements of students. |
| (*Ramadoss & Wang, 2012*) | 2012 | Grammar Grabber | Grammar | School | Frequencies | Students recognized that applications provide a safe environment where they learn by making mistakes. |
| (*Ishaq et al., 2020b*) | 2020 | Literacy and Numeracy Drive | Vocabulary learning and Reading comprehension | School | Mean, SD | The finding showed that the application is not efficient in teaching and learning English (specifically English Comprehension) at grade 03 due to poor design and usability, content, and assessment issues students face. Fun based learning (Serious Game) demanded by the students and teachers. |
| **Gamified Apps (Serious Game)** | | | | | | |
| (*Sandberg, Maris & Geus, 2011*) | 2011 | MEL Application | Vocabulary learning | School | Paired T-Test | The results revealed that students are motivated to use the application in their spare time. |
| (*Holden & Sykes, 2011*) | 2011 | Mentira | Spanish language learning | University | Interview | The design and use of augmented reality games on mobile devices to motivate place-based education is emerging. |
| (*Sandberg, Maris & Hoogendoorn, 2014*) | 2014 | Mobile English Learning2 (MEL2) (Game features added) | Vocabulary learning | School | MANOVA, ANCOVA, and Correlation | The results presented indicate that the students who worked with MEL-enhanced outperform the children who used MEL-original. |
| (*Wu, 2018*) | 2018 | English Vocabulary Practice System Game | Vocabulary learning | University | Mean, SD, t-test | The game-based vocabulary practice system had higher learning effectiveness for students with positive feedback. |
| (*Amer, 2014*) | 2014 | Idiomobile (Game for idioms) | Idioms | University | Mean, Correlation | Results showed that participants have positive attitudes toward the use of mobile technology in language learning. |
| (*Tsai et al., 2016*) | 2017 | Game-based Happy English Learning System | Vocabulary, Grammar, Sentence Structure | School | Mean, SD, Paired-sample t-test, and multiple linear regression | The results confirmed the positive effects of encouraging students' achievement and motivation for learning English. |
| (*Kohnke, Zhang & Zou, 2019*) | 2019 | Excel@EnglishPolyU (Alphabet vs. Aliens and Books vs. Brains@PolyU) | Vocabulary learning | University | Frequencies | The study demonstrates that mobile gamified educational programs are a productive avenue for students to expand their business vocabulary. |

| Ref. | Year | Applications | Purpose | Level | Evaluation | Result |
|---|---|---|---|---|---|---|
| (Chen, Liu & Huang, 2019) | 2019 | PHONE Words | Vocabulary learning | University | ANOVA, Frequencies | The results showed that the performance in vocabulary acquisition and retaining by the experimental group was significantly higher. |
| (Fisser, Voogt & Bom, 2012) | 2013 | Word Score | Vocabulary learning | School | Mean, SD, t-test | The results revealed teachers' and students' positive experiences and significant performance noted who played outside the class time. |
| (Önal, Çevik & Şenol, 2019) | 2019 | SOS Table | Tenses, Words, Sentences | University | t-test, thematic analysis | Results concluded that SOS Table is effective and efficient. |

Quan, 2016) (Makoe & Shandu, 2018) (Klimova & Polakova, 2020), pronunciation (Liakin, Cardoso & Liakina, 2014), writing, reading, listening and speaking (Chang et al., 2013), listening comprehension (Çakmak & Erçetin, 2017), grammar (Chu, Wang & Wang, 2019) and reading comprehension (Ishaq et al., 2020b). The results presented by studies for the applications of pedagogy, reading, pronunciation, writing, listening, speaking, grammar outperformed the control group. It may be concluded that mobile applications for vocabulary learning have a positive impact on language learning in which pupils from the experimental group outperformed the control group except (Klimova & Polakova, 2020) (Ishaq et al., 2020b) where the students and teachers mentioned they are not satisfied with the design, content, assessment methods adopted in the applications besides pronunciation support and teachers' notifications were also missing.

Gamified applications are proposed for vocabulary learning (Sandberg, Maris & Geus, 2011) (Sandberg, Maris & Hoogendoorn, 2014) (Wu, 2018) (Chen, Liu & Huang, 2019) (Fisser, Voogt & Bom, 2012), Spanish language learning (Holden & Sykes, 2011), Idioms (Amer, 2014), Vocabulary, Grammar, and Sentence Structure (Tsai et al., 2016), and Tenses, Words, Sentences (Önal, Çevik & Şenol, 2019). The results presented by studies for gamified applications of vocabulary learning, Idioms, grammar, sentence structure, tenses, and words outperformed the control groups with much engagement, interest, and positive feedback. Although mobile applications are effective, gamified applications are more effective with enhanced interest and engagement, resulting in positive learning outcomes (Ishaq et al., 2020a; Ishaq et al., 2020b; Ishaq et al., 2019; Ishaq et al., 2020d). Furthermore, it may be concluded that gamified applications are trending for all the subjects at all levels of education to get significant learning performance (Ishaq et al., 2020d) (Ishaq et al., 2020c; Dichev & Dicheva, 2017) (Zin & Yue, 2013; Zin, Jaafar & Yue, 2009). Furthermore, (Ishaq et al., 2020b) recommended gamified application in the public sector school to effectively learn the English language (specifically reading comprehension at primary level) after addressing all the stakeholders' issues.

## DISCUSSION AND FUTURE DIRECTIONS

This section summarizes and discusses the results related to the systematic literature review.

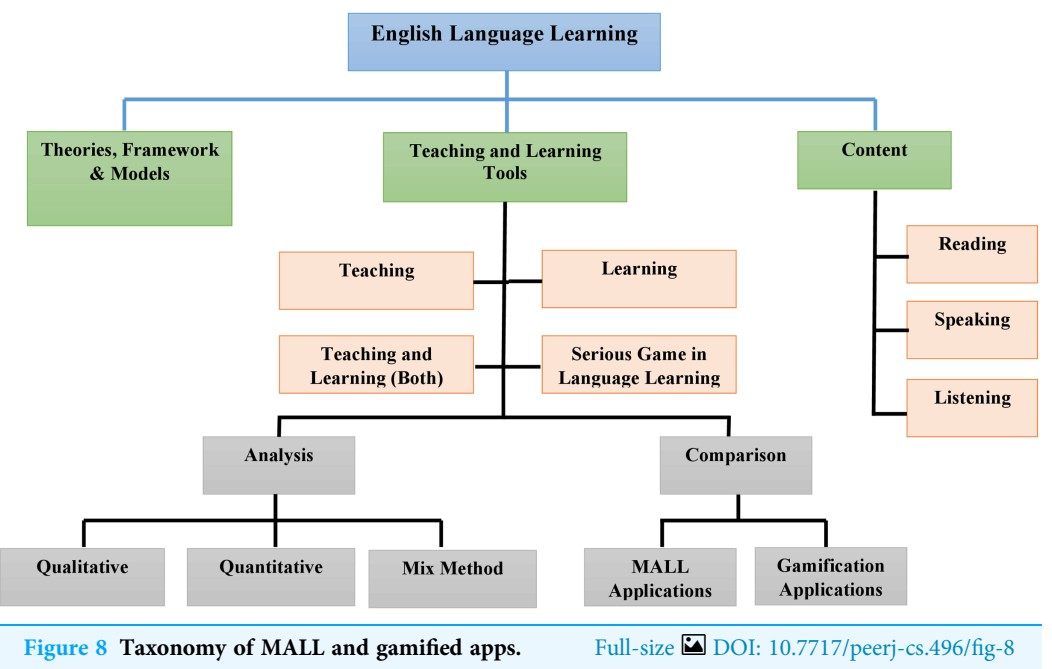

**Figure 8** Taxonomy of MALL and gamified apps.     

## Taxonomic hierarchy

In this comprehensive literature review, the aim was to investigate current MALL information and its application with 63 selected papers. To achieve this objective, a taxonomic hierarchy of finalized studies shown in Fig. 8 was established and examined trends and perspectives on adapted/developed frameworks, content, and teaching and learning tools. However, these dimensions were further broken down into several sub-levels that illustrated each area's scope with its role in enhancing the efficiency of language learners.

## Future directions

It was suggested that MALL specific theories or models might be developed because the models used from fields other than MALL was to be adopted or adapted as all components of the models used in the studies could not be applied to measure the constructs. The researchers used varying theories or models to develop a theoretical framework of the respective studies, and this practice was too time-consuming, and the models were minimum valid. The researcher confronted the same issue while measuring usability, usefulness, effectiveness, ease of use, user-friendliness, and user behavior. To ensure the effectiveness of learning content used in mobile and gamified applications, it was suggested that duly approved content material, i.e., written or pictorial by the concerned authorities, was observed from the literature. Whereas specified and authentic learning content was missing, and desired learning outcomes were not achieved. More work is needed concerning teachers' professional development and exploration of their competencies in the field of MALL. The teacher-student relationship was inevitable regarding the teaching-learning process because it was the only source of incorporating various teaching methods and techniques to make learning effective (*Omer, Farooq & Abid, 2020*). In this

regard, to compete for the world, teachers must link up with technological devices and gamified applications to engage their students in learning and entertainment to achieve a high learning outcome (*Isa et al., 2019*).

The applications used for learning did not fulfill the requirements related to usability, usefulness, effectiveness, ease of use, user-friendliness, and user behavior, so it is proposed to develop gamified applications (serious game) having all features supporting the above constructs. From the literature review, a gamified application for English language learning at the primary level was lacking. Students of primary class levels loved to play more than indulging in books. A conventional environment made the students passive in the class and lessened their motivation level.

During the literature review, research work regarding development of gamified application for reading comprehension was lacking. No such conceptual model gamification has been developed which may be used for language learning with reference to cultural context. Assessment of grammar or vocabulary in MALL applications were found during literature review but were not as such advanced enough that might perform personalized assessment of students' learning outcomes. Same was found in case of reward system and entertainment elements.

Therefore, gamified learning applications may be designed for primary grade students to make language learning more enjoyable, entertaining, attractive, and engaging them to achieve high learning outcomes effectively.

## CONCLUSION

This systematic literature review aimed to understand research patterns in MALL to learn the English language through mobile or gamified applications, approaches, and frameworks/models developed, or adopted. A comprehensive analysis of literature was undertaken to ensure a detailed discussion of the problems and their remedies. It was searched with as many known terminologies associated with MALL and then analyzed the results accordingly. The search was ended in August 2020, which would not have comprised studies that were carried out after the date. The Web of Science core collection was analyzed, and 63 out of 57,364 publications were selected.

The findings shown that nearly every selected article was published in a recognized journal, whereas only single research at a conference. The two primary forms of study adopted in these studies were "Solution Proposal" and "evaluation research". The majority of the chosen researches were evidence-based and could lead to the full advantages of MALL to teachers and students. MALL's most frequent key aspects were language learning strategies and evaluation of students' results. In contrast, MALL specific frameworks and theories, approved content were less addressed aspect of MALL.

The deficiencies in SLR related explicitly were research technique, incorrect data collection, or misclassification. However, with separate keywords from the Web of Science core collection repository, the research approach minimized the possibility of selection error. External concerns were addressed by implementing specific inclusion/exclusion guidelines, and two independent experts were requested to evaluate all extractions.

For future research on MALL, more attention might be paid to primary or secondary school students and teachers, approved curricula design for English subject, and tools' design, particularly serious games. Further evaluation research might be conducted to analyze existing MALL content.

## ACKNOWLEDGEMENTS

This study is a part of my doctoral thesis work at the Faculty of Information Science and Technology, Universiti Kebangsaan Malaysia, Malaysia. I would like to express my deepest gratitude to my advisors, Prof. Dr. Nor Azan Mat Zin, Dr. Fadhilah Rosdi, and Prof. Dr. Adnan Abid. I also wish to express my sincere thanks to all the co-authors who have contributed to this work.

### Funding

This work was supported by the Universiti Kebangsaan Malaysia under the Grand Challenge Fund (Grant number DCP-2017-007/2). The funders had no role in study design, data collection and analysis, decision to publish, or preparation of the manuscript.

### Grant Disclosures

The following grant information was disclosed by the authors:
Universiti Kebangsaan Malaysia: DCP-2017-007/2.

### Competing Interests

Adnan Abid is an Academic Editor for PeerJ Computer Science.

### Author Contributions

- Kashif Ishaq conceived and designed the experiments, performed the experiments, analyzed the data, performed the computation work, prepared figures and/or tables, authored or reviewed drafts of the paper, and approved the final draft.
- Nor Azan Mat Zin conceived and designed the experiments, analyzed the data, authored or reviewed drafts of the paper, and approved the final draft.
- Fadhilah Rosdi conceived and designed the experiments, analyzed the data, authored or reviewed drafts of the paper, and approved the final draft.
- Muhammad Jehanghir performed the experiments, analyzed the data, performed the computation work, prepared figures and/or tables, and approved the final draft.
- Samia Ishaq performed the experiments, analyzed the data, performed the computation work, prepared figures and/or tables, and approved the final draft.
- Adnan Abid conceived and designed the experiments, analyzed the data, performed the computation work, authored or reviewed drafts of the paper, and approved the final draft.

## Data Availability

This is a Systematic Literature Review and does not depend on raw data or code.

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
