# Peer review of "Mobile-assisted and gamification-based language learning: a systematic literature review"

_PeerJ Computer Science, doi:10.7717/peerj-cs.496_

## Round 0.1 · original submission · Minor Revisions

Dear Authors,

We have received reports from the reviewers. The paper has merits but needs minor revision before publication. I recommend revising the paper carefully to accommodate all the reviewers' recommendations.

Please add more recent articles as a reference in all the sections to make the article up-to-date.

Please adhere to the journal author's guidelines fully.

Yousaf Bin Zikria
Academic Editor

Reviewer 1 ·

Basic reporting

no comment

Experimental design

no comment

Validity of the findings

no comment

Additional comments

The article presents a systematic literature review of mobile assisted and gamification based
language learning. The article covers different important aspects of the study including teaching
and learning approaches, contents and curriculum, and evaluation measures. The discussion is
well structured and well presented.
The selection of articles follows a systematic and well-defined methodology. A reasonable
number of recent articles published in high quality venues have been selected for this study.
However, the abstract and introduction of the article need to be improved. In abstract the authors
should provide the major contributions preferably in quantitative terms. Similarly, the motivation
and justification along with major contributions needs to be clearly presented in the introduction
section.
At several places, the authors have provided a list of references together during the discussion. It
will be more appropriate to discuss relevant articles while discussing their relevance as well as
the differences in their contributions (instead of placing them under one discussion point). Some
more details should be shared to highlight the differences between simple mobile applications
and serious games.
Lastly, the language quality needs to be improved.
As a whole, the article presents useful and valuable contribution which can be helpful for the
researchers working in this area, and can be accepted for publication after addressing the given
comments.

·

Basic reporting

In this article, the authors have provided a systematic literature review on an interesting topic of learning languages using mobile phones that involves mobile applications as well as mobile games. It presents a review of more than 60 good-quality articles.
The whole discussion has been structured in a very interesting flow that starts with theories, frameworks, and models. Then tools and approaches used for teaching and learning with and without gamification have been discussed. Similarly, contents used for teaching and learning have also been discussed. While different aspects of the evaluation of such approaches are also present. A useful taxonomy and interesting future directions have also been proposed. The article is generally easy to follow, covers an ample amount of background studies and state of the art, and is structured in a good way.

Experimental design

I believe the article falls within the domain of the journal. The methodology provided herein seems comprehensive and consistent throughout. The article is overall organized in a well-organized way with coherence among the paragraphs. The strength of the article is its vast coverage with the help of good quality literature.

Validity of the findings

The results of the paper are encouraging and the conclusion supports the findings of the papers in general.

Additional comments

There are still some minor issues with the article that need to be fixed before its acceptance.
1-It will be useful if the authors could write the article in an impersonal manner and avoid using the pronouns (we and us etc). The quality of some figures needs to be improved.
2-During the synthesis and discussion, there is a need to enrich discussion while explaining the obtained statistics with reasoning, for instance, in the Taxonomic Hierarchy sub-section there is a need to reason with the help of statistics.
3-Similarly, future directions need to be presented in a more structured manner. At different places, they seem to be pretty arbitrary.
4-At a couple of places, there are long streaks of references, which look unusual. Particularly, in the Tools used for Analysis sub-section, there is a long list of references that takes almost two lines. This needs to be improved.
5-Language quality in Abstract, Introduction, and Future Directions needs to be improved.

---

## Round 0.2 · accepted · Accept

Thank you for revising the paper based on the reviewers comments.The reviewers are happy with the revised manuscript.

Reviewer 1 ·

Basic reporting

The paper presents the topic very well now and accumulates suggested changes now.

Experimental design

The recommended changes in technical content are addresses.

Validity of the findings

The presentation of results is better now.

Additional comments

The authors have successfully incorporated all the suggested changes.
Overall, the paper seems in better form as it accumulates the topic and content under it very well now.

·

Basic reporting

I have checked the revised version. It seems the authors have addressed all of my comments of basic nature.

Experimental design

The minor design issues have been resolved.

Validity of the findings

I don't have further comments on the findings.

Additional comments

I have checked the revised version. It seems the authors have addressed all of my comments and I don't have any further comments. Hence, the paper could be accepted in the current form.